# FROM ATTENTION TO PREDICTION MAPS: PER-CLASS GRADIENT-FREE TRANSFORMER EXPLANATIONS

## ABSTRACT

The Vision Transformer (ViT) has become a standard model architecture in computer vision, especially for classification tasks. As such, explaining ViT predictions has attracted significant research efforts in recent years. Many methods rely on attention maps, which highlight *where* in the image the network directs its attention. In this paper, we introduce Prediction Maps – a novel explanation method that complements attention maps by revealing *what* the network sees. Prediction maps visualize how each patch token within a given layer is associated with each possible class. This is done by utilizing the classification head at the output of the network, originally trained to be fed with the class token at the last layer. Specifically, to obtain the prediction map of a particular layer, we apply the classification head to every patch token within that layer. We show that prediction maps provide complementary information to attention maps and illustrate that combining them leads to state-of-the-art explainability performance. Furthermore, since our proposed method is neither gradient- nor perturbation-based, it offers superior computational and memory efficiency compared to competing methods. To the best of our knowledge, ours is the first explainability method for ViTs that is both class-specific and gradient-free

## 1 INTRODUCTION

Following their introduction in the context of language models, transformers (Vaswani et al., 2017) have become the neural architecture of choice across diverse machine learning domains. They have been adopted e.g. in graph neural networks (Dwivedi & Bresson, 2020; Yun et al., 2019) and for point-cloud analysis (Qin et al., 2022; Zhao et al., 2021), and have also been extended to a wide range of vision tasks, including detection (Carion et al., 2020; Li et al., 2022; Misra et al., 2021), classification (Dosovitskiy et al., 2020; Zhao et al., 2021), segmentation (Kirillov et al., 2023; Zheng et al., 2021), and image generation (Li et al., 2019; Touvron et al., 2023).

Given their pervasive dominance, significant research efforts have been devoted to understanding how transformers process their inputs, as well as to explaining their predictions, with particular focus given to the vision transformer (ViT) architecture Abnar & Zuidema (2020); Chefer et al. (2021a;b); Liu et al. (2021a); Mohankumar et al. (2020); Wu et al. (2024). Many methods rely on the attention maps within the model to provide explanations for its predictions (Abnar & Zuidema, 2020; Chefer et al., 2021b). Raw attention maps are appealing because (i) they are calculated as part of the forward-pass of the network and thus do not require any additional computations to extract, and (ii) they provide a glimpse into how the model constructs its prediction. However, attention maps only offer insights into *where* in the input image the network focuses its attention, and do not visualize *what* the network "perceives" within each region of the image. In other words, they do not indicate the extent to which each region is associated with a specific class.

A common approach for providing more informative explanations, is to seek for heatmaps that visualize the contribution of each patch in the input image to each possible class prediction. Methods that generate such visualizations can be broadly categorized as *perturbation based* (Carter et al., 2019; Fong et al., 2019; Fong & Vedaldi, 2017; Lundberg & Lee, 2017; Petsiuk et al., 2018; Ribeiro et al., 2016) or *gradient based* (Bach et al., 2015; Chefer et al., 2021a;b; Selvaraju et al., 2017; Sundararajan et al., 2017). Perturbation-based methods treat the model as a black box, inspecting how its output changes in response to small perturbations to its input. Gradient-based methods

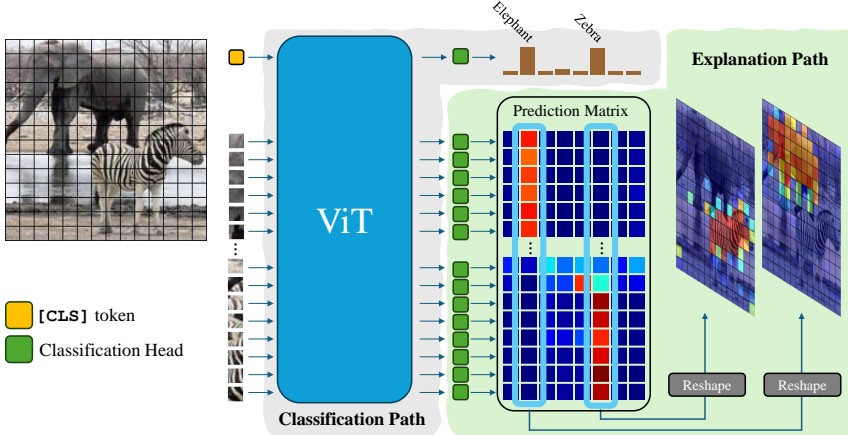

Figure 1: **Prediction map construction overview.** Applying the classification head on all patch tokens yields a class-specific per token classification. Although it was originally trained on the class token, the predictions obtained from other patches are satisfying.

perform a backward pass to accumulate gradients propagated through the entire network. However, both approaches are associated with heavy computational costs, and do not directly shed light on how the model processes data in its forward pass.

In this paper, we introduce *Prediction Maps* – a lightweight gradient-free explainability method that is as simple and fast as extracting attention maps and has the per-class expressiveness of the sophisticated perturbation- and gradient-based methods. Our approach relies on the observation that when the classification head of a pretrained ViT, which is normally applied to the class-token at the last layer, is fed with any other (patch) token, it tends to output valid predictions. Surprisingly, we find that this is true not only for the patch tokens at the last layer, but also for the patch tokens at all other layers. This observation allows us to construct a heatmap for any desired class, as demonstrated in Fig. 1, as well as to visualize how the localization of concepts evolves throughout the layers.

Prediction maps provide complementary information to attention maps. Therefore, their joint inspection can yield a more comprehensive explanation than each of them alone, enabling to expose the root cause of incorrect predictions. To illustrate this, we show in Fig. 2 two failure cases of a ViT on images from the ImageNet-R dataset (Hendrycks et al., 2021a). In the first, the model misclassifies a lemon as tray. Here, the attention and prediction maps reveal that the lemon is actually recognized correctly, but is not attended by the model. In the second example, the model misclassifies a violin as a hair slide. Here, the attention and prediction maps reveal that the violin is well attended, yet not recognized, possibly because of the pencil strokes that look like hair.

To obtain a single map that combines the explainability power of both approaches, we propose to unify them into a visualization which we term *PredicAtt*. We use the fact that the correlation between an attention map and a prediction map indicates how much that attention map contributes to the classification. We therefore construct a weighted combination of the attention maps from all heads and layers according to their similarity with the prediction map. We then compute the per-element product between this weighted attention map and the prediction map to obtain our combined class-specific map. This enables the analysis of how each layer and head recognizes each class.

Our main contributions can be summarized as follows:

1. We introduce *prediction maps* for explainability as a complementary component for the well studied attention maps. Prediction maps provide a per-token measurement of what the network perceives. Additionally, we propose a simple way to combine attention maps and prediction maps, termed *PredicAtt*.

2. Our method is gradient-free and perturbation-free; hence, it is efficient in terms of runtime and memory consumption. To the best of our knowledge, it is the first explainable method for ViT that is both class-specific and gradient-free.

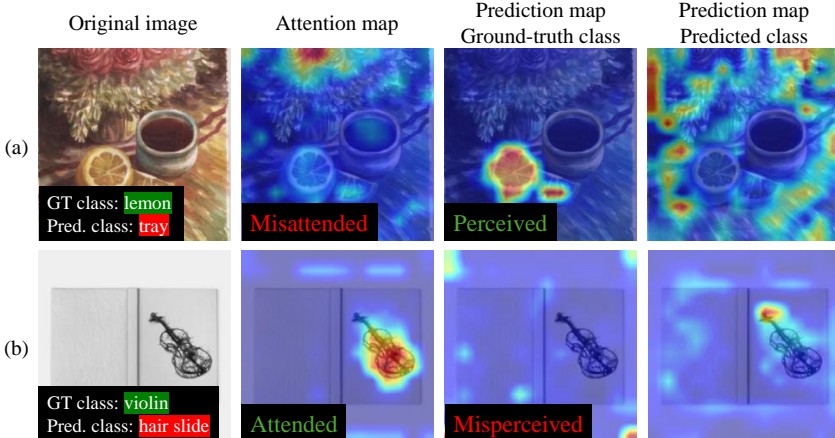

Figure 2: **Gradient-free analysis of failure cases.** (a) The lemon object is perceived accurately, but ViT attends elsewhere. (b) The object is accurately attended; nevertheless, the network mispredicts it, possibly due to the pencil strokes that look like hair. Analyzing ViT through the lens of prediction maps, which complement attention maps, provides interesting insights into the root cause of failure cases. Prediction map of the predicted class is given for completeness.

3. We show that the correlation between prediction maps and attention maps offers insights into how the ViT processes data at a head granularity level. Combining prediction maps with attention maps provides per-class explanations, enhancing interpretability.

4. Our approach achieves state-of-the-art results on perturbation and segmentation tests utilizing ViT-B and ViT-L on the ImageNet dataset (Russakovsky et al., 2015).

## 2 RELATED WORK

Explainable AI has been widely studied across various domains utilizing deep neural networks, including NLP (Li et al., 2015; Chefer et al., 2021a), speech (Bharadhwaj, 2018; Kumar et al., 2021), point cloud analysis (Levi & Gilboa, 2024; Zheng et al., 2019), and graph neural networks (Ying et al., 2019; Yuan et al., 2021). In the realm of image classifiers, many works aim at producing a heatmap that highlights which regions in the input image affect the classifier's prediction the most.

**Explainability for arbitrary architectures.** Generic methods for explaining the predictions of image classification models can be roughly categorized as gradient-based or perturbation-based. Gradient based methods use backpropagation in various ways. For example, GradCAM (Selvaraju et al., 2017) computes the gradient of the score for any queried class with respect to the last pooling layer. Layer-wise Relevance Propagation (LRP) (Bach et al., 2015) propagates relevance values, which are based on gradients, through the entire network. Integrated Gradients (Sundararajan et al., 2017) accumulates the gradients on a path from a baseline image to the scrutinized one. These approaches, however, are computationally demanding. Moreover, they do not shed light on how the computations within the network's forward pass lead to its prediction. Perturbation-based methods are model-agnostic, treating the model as a black box and inspecting how its output changes in response to small perturbations to its input. LIME (Ribeiro et al., 2016) learns a linear model based on perturbations of the input sample. SHAP (Lundberg & Lee, 2017) translates the concept of Shapley values to model explainability. RISE (Petsiuk et al., 2018) estimates pixel importance based on probing the model with randomly masked versions of the input image. Fong & Vedaldi (2017) optimize perturbations on the input data to identify the smallest most influential regions that significantly affect the model's output. Extreme Image Transformations (Malik et al., 2023) is a more computationally efficient approach, analyzing how significant alterations to the input influence the model's predictions. Carter et al. (2019) identify minimal subsets of features whose observed values alone suffice for the same decision to be reached. Despite being applicable to any model, many of these methods suffer from extreme computational costs.

**Explainability for convolutional networks.** Class Activation Mapping (CAM) (Zhou et al., 2016) generates explainable maps for convolutional neural networks (CNNs) by leveraging the weighted sum of the feature maps from the final convolutional layer, guided by class-specific weights from the fully connected layer. Our prediction maps can be viewed as an adaptation and extension of CAM to ViTs. While CAM in CNNs can only be applied to the last layer due to differing channel dimensions across earlier layers, prediction maps can be applied to any layer in ViTs. Furthermore, we demonstrate how to combine prediction maps with attention maps to enhance the explainability.

**Explainability for transformers.** With the widespread adoption of transformers (Vaswani et al., 2017), many explainability methods emerged particularly for those architectures. Most methods make use of the self-attention mechanism, utilizing attention maps for explainability (Abnar & Zuidema, 2020; Chefer et al., 2021a;b; Liu et al., 2021a; Mohankumar et al., 2020; Wu et al., 2024). While early works used raw attention maps for explanation, subsequent approaches explored more sophisticated techniques, such as rolling out attention information from all layers (Abnar & Zuidema, 2020). Voita et al. (2019) applied LRP to transformers, focusing only on attention head relevance. Chefer et al. (2021b) further adapted LRP, allowing propagating relevance scores through all layers, while Chefer et al. (2021a) generalized the method to models with cross-attention modules. Wu et al. (2024) incorporated the influence of token transformations into their explainable method, particularly focusing on changes in the tokens' norms and directions in each attention block.

## 3 BACKGROUND

In this section we provide a brief overview of the Vision Transformer (ViT) architecture, mainly to set notations. A more comprehensive description can be found in Dosovitskiy et al. (2020).

A ViT consists of a stack of $L$ transformer encoder layers, each comprising a multi-head-self-attention (MHSA) module and a feed-forward network (FFN) block with skip connections (He et al., 2016). The input to each layer is a sequence of tokens, each corresponding to a distinct patch within the input image, along with an additional special token for classification called the [CLS] token (Devlin et al., 2018). The [CLS] token is designed to aggregate information from all other tokens to enable classification based on that token alone. Therefore, at the last layer, the [CLS] token is fed into a classification head, which outputs the predicted class.

We denote by $x^{(l)} \in \mathbb{R}^{(N+1) \times d}$ the output of layer $l$, where $N$ is the number of regular (patch) tokens (excluding the [CLS] token) and $d$ is the embedding size. Thus, $x^{(0)}$ is the network's input embedding and $x^{(L)}$ is the output of the last layer. At the output of layer $l$, we denote the [CLS] token by $x_{\text{CLS}}^{(l)} \in \mathbb{R}^d$ and the $i$th patch token by $x_i^{(l)} \in \mathbb{R}^d$, for any $i \in \{1, \cdots, N\}$. Each transformer layer contains a MHSA block with $H$ heads, each outputting a vector of dimension $d_H$ per token. These $H$ vectors are concatenated and linearly transformed back to dimension $d$. The $h$th head within the $l$th layer applies three linear transformations to $x^{(l)}$, with matrices $\{Q^{(l,h)}, K^{(l,h)}, V^{(l,h)}\} \in \mathbb{R}^{(N+1) \times d_H}$. The results of these transformations are the queries, keys and values, respectively. An *attention matrix* is then constructed as

$$A^{(l,h)} = \text{softmax}\left(\frac{Q^{(l,h)} K^{(l,h)^\top}}{\sqrt{d_H}}\right) \in \mathbb{R}^{(N+1) \times (N+1)}, \tag{1}$$

where the softmax operates on the rows of its matrix argument. The *attention map* $A_{\text{CLS}}^{(l,h)} \in \mathbb{R}^N$ is the row of $A^{(l,h)}$ corresponding to the [CLS] token, excluding the entry of the attention between the [CLS] token and itself. The attention map captures the relation between the [CLS] token and each patch token. Reshaping it back to the image dimensions may serve to explain the model's prediction.

## 4 LIMITATIONS AND CHALLENGES IN USING ATTENTION MAPS FOR EXPLAINABILITY

The question of whether raw attention maps provide informative explanations has attracted a lot of debate (Bibal et al., 2022; Jain & Wallace, 2019; Wiegreffe & Pinter, 2019). Here, we list several of the obvious limitations and challenges associated with using them for explainability.

Perhaps the most fundamental limitation of attention maps is their class-agnostic nature. Since there is no attention map per class, they do not reveal to what extent each token "recognizes" each possible class. Such a visualization is often highly desirable for explaining a model's prediction. For example, in cases of incorrect classification, it is informative to inspect the extent to which each token "recognizes" the true class vs. the predicted class (see Fig. 2). Additionally, when the input image contains multiple objects, it is desirable to visualize how each token "recognizes" each of the classes present in the image.

Recent work, such as the Multi-class Token Transformer (MCTformer) (Xu et al., 2022), demonstrates an alternative approach to obtaining class-specific attention maps by introducing multiple class tokens, each representing a distinct class. This approach is both gradient-free and class-specific, addressing the limitations of conventional attention maps. However, unlike MCTformer, our method achieves class-specificity without requiring architectural changes or additional class tokens, offering a more generalizable solution for visualizing class-discriminative behavior in standard Vision Transformers.

Another challenge in using attention maps for explainability relates to the difficulty in determining how to select or combine attention maps from the different layers and heads. Some works use the attention maps of the last layer, assuming they capture the most high-level semantics (Caron et al., 2021). Other methods attempt to find a better combination of attention maps (Abnar & Zuidema, 2020; Chefer et al., 2021a;b). For example, the Rollout (Abnar & Zuidema, 2020) method uses $(I + A^{(1)}) \cdot (I + A^{(2)}) \ldots (I + A^{(L)})$, where $I$ is the identity matrix. Yet, determining the optimal combination of attentions from different layers is still an active area of research. As for selecting the attention maps from the different heads within each layer, some methods heuristically opt to average them (Abnar & Zuidema, 2020; Tang et al., 2018; Voita et al., 2018). However, it has been demonstrated that different heads may capture different semantics (Voita et al., 2019), even within the same layer. Recently, Darcet et al. (2023) showed that the network sometimes encodes global information in some of the tokens. In such cases, the corresponding entries in the attention map may no longer represent the original semantic meanings of the tokens, and the effect of averaging across attention heads remains unclear.

Finally, relying solely on attention maps may fail in the face of dataset biases, where the network can exploit shortcuts or spurious cues (Geirhos et al., 2020; Hendrycks et al., 2021b), resulting in the highlighting of non-discriminatory regions. For instance, it has been shown that the attention maps of a cow in a pasture image tend to over-attend to the grass rather than the cow itself. This is despite the fact that the grass is not the most important region for constructing the network's prediction, as revealed by perturbation tests (Chefer et al., 2022). That is, the network persists to classify the cow correctly even when the grass patches are masked.

## 5    PREDICTION MAPS

Attention maps capture the interactions between the queries and keys, indicating *where the network looks*. Yet, they do not provide insights into *what the network sees*, which is encoded in the values within the self-attention mechanism. To address this gap, we introduce the concept of *Prediction Maps* which decode the data in the values into a human-understandable visualization. Recall that the classification head, $\nu : \mathbb{R}^d \to \mathbb{R}^C$, is normally fed with $x_{\text{CLS}}^{(L)}$, where $C$ is the number of classes, to output the prediction $y = \nu(x_{\text{CLS}}^{(L)})$. Here, we propose to feed this head with the patch tokens, rather than the [CLS] token. Specifically, we define the *prediction matrix* of layer $l$ as

$$\Psi^{(l)} := \begin{bmatrix} \nu\left(x_1^{(l)}\right) & \nu\left(x_2^{(l)}\right) & \ldots & \nu\left(x_N^{(l)}\right) \end{bmatrix}^\top \in \mathbb{R}^{N \times C}. \tag{2}$$

This matrix provides a classification result based on each token within layer $l$ separately (see Fig. 1). The $i$th row of this matrix is the probability vector associated with predicting all possible classes based on the $i$th token. The $c$th column, denoted as $\Psi^{(l)}(c) \in \mathbb{R}^N$, forms the *prediction map* of class $c$. This map contains the prediction scores of class $c$ obtained for each patch token.

Surprisingly, despite the fact that the classification head has been originally trained to operate on the [CLS] token, we illustrate empirically that feeding it with patch tokens yields sensible results (see Fig. 3). This can be attributed to the fact that the [CLS] token and the patch tokens undergo the same processing within the model. Specifically, every token at the output of an attention layer

Figure 3: **Prediction map per layer.** Surprisingly, the classification head of a pretrained ViT works well on all tokens from all layers, albeit being originally trained to predict the class only based on the class-token of the last layer. The deeper the layer, the more it captures high-level semantics. We hypothesize that the last layer's prediction is worse than that of the penultimate layer, because the patch tokens at its output did not participate in the training of the model.

(including the `[CLS]` token) is a linear combination of (the Values of) all the tokens at the input of that layer. Furthermore, the weights of this linear combination are computed using the Queries and Keys of the tokens at the input, which are computed in the exact same manner for all tokens (namely, the same set of matrices is applied to each token to compute its $Q$, $K$, and $V$). Therefore, there is perfect symmetry in how the tokens are treated, which causes all tokens (including the `[CLS]` token) to encode information in the same manner. This is why applying the classification head to patch tokens provides sensible classification logits, just like when applying it to the `[CLS]` token. We note that the object detection method of Minderer et al. (2022) proposed to train a classifier based on patch tokens of the last ViT layer. This is in sharp contrast to our method, which uses the pretrained classification head 'as is' and also applies it to tokens from earlier layers.

The prediction map mechanism offers several advantages for explainability. First, it allows to visualize *what the network sees* in each region of the image. Second, it is computationally efficient, as it uses the tokens that are anyway computed during the forward pass of the network. Third, it sheds light on how the network constructs its prediction, as it does not rely on indirect measures based on gradients. Lastly, it provides class-specific maps. As opposed to attention maps, prediction map can be constructed for any desired class. For example, for an ImageNet classifier, it is possible to construct 1,000 prediction maps for each head within each layer, one prediction map per class.

## 5.1 PREDICATT: COMBINING PREDICTION MAPS WITH ATTENTION MAPS

While prediction maps offer advantages over attention maps, they lack information regarding where the network attends. To incorporate such information, we propose to integrate prediction maps and attention maps into a unified visualization, which we term *PredicAtt*. As we show in Sec. 6, this leads to state-of-the-art explainability results.

To generate a combined map for a specific class $c$, we follow a two-step process. First, we construct a combined attention map for that class, $\tilde{A}_{\text{CLS}}(c) \in \mathbb{R}^N$, by computing the weighted sum of attention maps across all heads and layers as

$$\tilde{A}_{\text{CLS}}(c) := \sum_{l=1}^{L} \sum_{h=1}^{H} \tilde{\alpha}_{l,h} A_{\text{CLS}}^{(l,h)}. \tag{3}$$

The coefficients in this weighted combination are computed based on the similarity between the attention maps and the prediction map of layer $i$,

$$\alpha_{l,h} = \left\langle A_{\text{CLS}}^{(l,h)}, \Psi^{(i)}(c) \right\rangle,$$
$$\{\tilde{\alpha}_{l,h}\} = \text{softmax}(\{\alpha_{l,h}\}), \tag{4}$$

where $\langle \cdot, \cdot \rangle$ is the standard inner-product $\langle a, b \rangle = a^\top b$. For details regarding an alternative similarity measure, please refer to Sec. A.2 of the supplementary. In the second step, we compute the per-element product between the class-specific weighted attention map and the class-specific prediction map, to yield

$$\text{PredicAtt}_i(c) := \tilde{A}_{\text{CLS}}(c) \odot \Psi^{(i)}(c) \ \in \mathbb{R}^N. \tag{5}$$

In principle, the prediction map $\Psi^{(i)}(c)$ in Eq. (5) can be taken from any layer $i$. However, as we show in Sec. 6.1, the last and second-to-last layers lead to the best results.

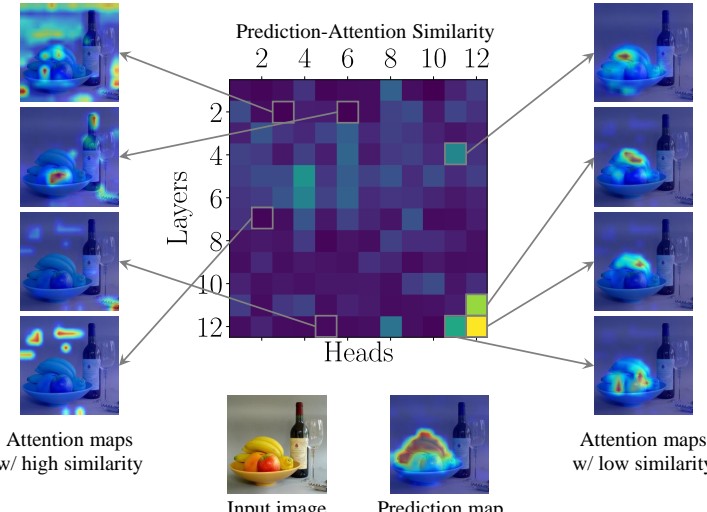

Figure 4: **Analyzing the contribution of attention maps through similarity to prediction maps.**
The similarity between an attention map and the prediction map reflects the degree to which this
attention map contributes to the prediction. Attention maps with high similarity to the prediction
map for class "banana", emphasize the banana object. In contrast, attention maps with low similarity
attend to the background, and are thus less informative for the classification. Interestingly, the least
and most contributing maps are not necessarily from the first and last layers, respectively. See the
supplementary for an example (Fig. 13).

Our approach is based on a per-head, rather than per-layer analysis and thus facilitates a more de-
tailed understanding of the network's operation. We claim that an attention map significant for
classification is one that yields a high dot product score with the prediction map. A higher dot
product score indicates that the network is attending to a relevant region of the image that is mostly
associated with a single class. On the other hand, if an attention map resonates over regions con-
taining multiple classes, then its impact on the classification is intuitively smaller, and accordingly,
it results in a lower dot product score. In other words, a high degree of similarity between the *what
the network sees* template and the *where to look* template signifies that the network has identified
meaningful features (see Fig. 4). We empirically verify this in Sec. 6.1 (Tab. 4, rows (i) and (ii)).

## 6 EXPERIMENTS

We evaluate our method on the task of explaining ViT classifier predictions. In Sections A.4 and A.5
of the supplementary, we further illustrate our approach on explaining the text model BERT (Devlin
et al., 2018), and the CLIP vision-language model (Radford et al., 2021).

We compare our method to several competing approaches, including the class-agnostic Raw At-
tention, Rollout (Abnar & Zuidema, 2020), LRP (Bach et al., 2015), and Partial LRP (Voita et al.,
2019) techniques, and the class-specific GradCAM (Selvaraju et al., 2017) and Transformer Attri-
bution (commonly referred to as TransAttr) (Chefer et al., 2021b) methods. Although the various
LRP method variants can generate explainable maps for each class, Chefer et al. (2021b) demon-
strate that, in practice, the visualizations across different classes are largely similar. Thus, we treat
these methods as class-agnostic. To reproduce the results of the competing methods, we follow the
implementation provided by Chefer et al. (2021b). In all our experiments, we use standard ViT-B/16
and ViT-L/16 models pretrained on ImageNet (Russakovsky et al., 2015). We evaluate two variants
of our PredicAtt method, constructed from the prediction maps of the last and second-to-last layers
(PredicAtt$_L$ and PredicAtt$_{L-1}$). Our implementation is based on the timm (Wightman, 2019).

Figure 5 shows qualitative comparisons on two representative images containing objects from dif-
ferent classes. As can be seen, our method better highlights the regions corresponding to each class.

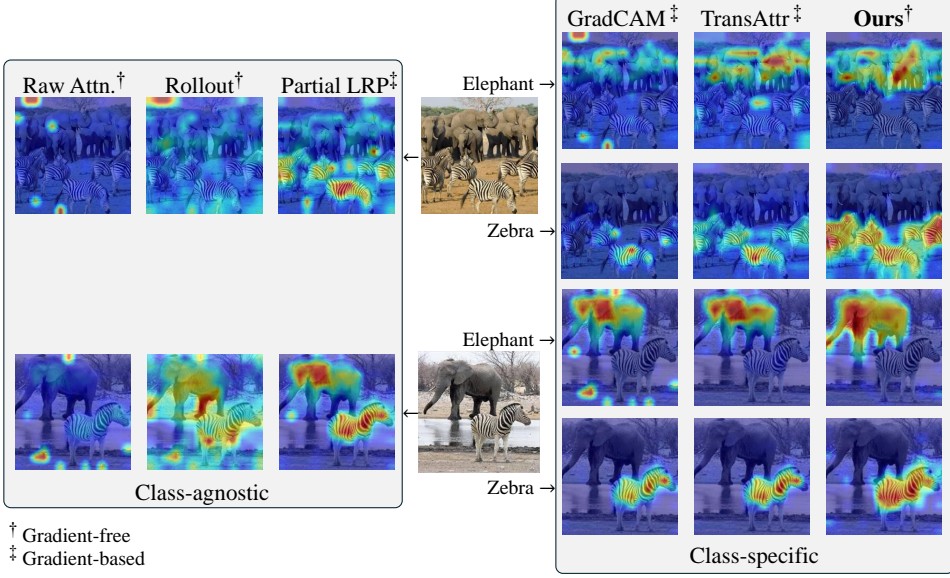

Figure 5: **Class-specific visualizations.** Our method, PredicAtt$_{11}$, captures a more coherent and compact region of the object. e.g. our method is the only one to highlight the elephant's trunk.

We provide more visual examples in the supplementary. As suggested by Chefer et al. (2021b), we quantify the quality of the explanations using two measures, as follows:

**Perturbation test.** In this test, patches in the input image are gradually masked based on their importance, while measuring the model's classification accuracy. This test has two variants; in the positive/negative version, pixels are masked in descending/ascending order of importance, leading to an expected sharp/gradual decline in accuracy. Both versions use the area under the curve (AUC) metric to quantify performance, scanning masking percentages of between 10% to 90% of the image. For the class-specific methods, we perform this experiment for both the predicted class and the target (ground-truth) class. Experiments are conducted on the ImageNet-Validation dataset (Russakovsky et al., 2015), which comprises 50,000 images from 1000 classes. The results are summarized in Tab. 1. As can be seen, both variants of our method achieve the best scores across all metrics.

| | | ViT-B/16 | | | | ViT-L/16 | | | |
|---|---|---|---|---|---|---|---|---|---|
| | | Negative | | Positive | | Negative | | Positive | |
| | Method | Pred. ↑ | Target ↑ | Pred. ↓ | Target ↓ | Pred. ↑ | Target ↑ | Pred. ↓ | Target ↓ |
| class-agnostic | Raw attention | 45.55 | - | 24.00 | - | 40.91 | - | 27.22 | - |
| | Rollout | 53.10 | - | 20.06 | - | 52.75 | - | 21.67 | - |
| | LRP | 43.69 | 43.69 | 41.94 | 41.94 | 40.28 | 40.27 | 39.99 | 39.99 |
| | Partial-LRP | 50.29 | 50.28 | 19.81 | 19.82 | 37.23 | 37.24 | 29.56 | 29.56 |
| class-specific | GradCAM | 41.53 | 42.03 | 34.05 | 33.54 | 46.99 | 47.07 | 45.16 | 45.06 |
| | TransAttr | 54.20 | 55.09 | 17.04 | 16.41 | 51.75 | 52.40 | 20.03 | 19.61 |
| | **PredicAtt$_{L-1}$ (Ours)** | 55.41 | 56.99 | **16.08** | **15.08** | **53.79** | **54.03** | 19.98 | 19.78 |
| | **PredicAtt$_L$ (Ours)** | **56.16** | **57.94** | 17.00 | 16.06 | 53.11 | 53.64 | **19.10** | **18.66** |

Table 1: **Perturbation test.** All methods are evaluated on the ImageNet validation set with the ViT-B/16 and ViT-L/16 models. Bold and underline mark the best and second best scores, respectively. The subscript in our method indicates the layer of the prediction map, where $L$ denotes the total number of layers in the model: 12 for ViT-B/16 and 24 for ViT-L/16.

**Segmentation test.** In this test, we use the explainability map generated from the predicted class to separate the foreground object from the background. To evaluate the performance of the segmentation, we employ three metrics: pixel accuracy, mean intersection-over-union (mIoU), and mean Average Precision (mAP), all computed based on the ground truth annotations. Our experiments are conducted on the ImageNet-Segmentation dataset (Guillaumin et al., 2014), comprising 4,276

| | | ViT-B/16 | | | ViT-L/16 | | |
|---|---|---|---|---|---|---|---|
| | Method | Pixel Acc.↑ | mIoU↑ | mAP↑ | Pixel Acc.↑ | mIoU↑ | mAP↑ |
| class-agnostic | Raw attention | 67.87 | 46.37 | 80.24 | 63.20 | 41.18 | 74.75 |
| | Rollout | 73.54 | 55.42 | 84.76 | 71.15 | 52.88 | 83.48 |
| | LRP | 50.77 | 32.64 | 55.90 | 49.81 | 31.87 | 54.73 |
| | Partial-LRP | 76.31 | 57.97 | 84.67 | 62.40 | 40.21 | 73.65 |
| class-specific | GradCAM | 65.91 | 41.31 | 71.60 | 68.49 | 39.73 | 63.30 |
| | TransAttr | 79.72 | 61.98 | 86.04 | 72.88 | 52.20 | 81.22 |
| | **PredicAtt$_{L-1}$ (Ours)** | **79.75** | **62.65** | **87.15** | 78.32 | 59.20 | 84.38 |
| | **PredicAtt$_{L}$ (Ours)** | 76.85 | 59.22 | 86.24 | **83.06** | **64.51** | **86.78** |

Table 2: **Segmentation Test.** All methods are evaluated on the ImageNet-Segmentation dataset with the ViT-B/16 and ViT-L/16 models. Bold and underline mark the best and second best scores.

| | | ViT-B/16 | | ViT-L/16 | |
|---|---|---|---|---|---|
| | Method | Runtime ↓ | Memory ↓ | Runtime ↓ | Memory ↓ |
| class-agnostic | Raw attention | 6 ms | 8 MiB | 11 ms | 11 MiB |
| | Rollout | 8 ms | 28 MiB | 19 ms | 68 MiB |
| | LRP | 158 ms | 685 MiB | 289 ms | 2006 MiB |
| | Partial LRP | 157 ms | 685 MiB | 285 ms | 2006 MiB |
| class-specific | GradCAM | 29 ms | 511 MiB | 42 ms | 1637 MiB |
| | TransAttr | 167 ms | 681 MiB | 341 ms | 2013 MiB |
| | **PredicAtt$_{L}$ (Ours)** | 7 ms | 68 MiB | 20 ms | 172 MiB |

Table 3: **Resource consumption.** All results are for a single image. Our method is as fast and lightweight as gradient-free methods, running an order of magnitude faster and consuming an order of magnitude less memory than class-specific gradient-based methods.

images from 445 classes, each annotated with manual segmentation delineating the object in every image. Results are presented in Tab. 2. On ViT-L/16, both variants of our method achieve the highest scores across all metrics. On ViT-B/16, our PredicAtt$_{L-1}$ variant achieves the highest scores on all metrics, while PredicAtt$_{L}$ achieves the second-highest mAP score and third-highest pixel accuracy.

**Resource consumption.** Table 3 reports the GPU memory usage and runtime of all methods. Memory consumption refers to the difference between the peak memory allocation recorded post-generation of the explainable map and the memory allocated prior to this process. It thus quantifies the memory overhead of each method while neutralizing the memory footprint of the model weights. Notably, our method, which does not require gradients or backward passes, exhibits significantly shorter runtime and reduced memory consumption, highlighting its efficiency compared to competing methods. Experiments were conducted on an NVIDIA GeForce RTX 2080 Ti GPU.

## 6.1 ABLATION STUDY

We next study the contribution of each component in our approach by evaluating the following variants: (i) using only an average attention map, (ii) using only a weighted attention map, and (iii) using only a prediction map. The segmentation and perturbation tests for these variants with the ViT-B/16 model are reported in Tab. 4. While using only attention maps or only prediction maps yields unsatisfactory results, combining them leads to state-of-the-art performance.

**Layer selection.** Prediction maps can be obtained from any ViT layer. To determine which layers are most appropriate for explaining the prediction, we repeat the perturbation and segmentation tests for all layers. In Fig. 6, we observe a trend of improvement in all metrics as we take the prediction maps from deeper layers in the network. However, there is a slight degradation in the last layer, particularly in the segmentation and positive perturbation metrics. This may be attributed to the fact that the patch tokens at the output of the last layer do not affect the model's output and are thus not

| | Segmentation test | | | Perturbation test | | | |
| --- | --- | --- | --- | --- | --- | --- | --- |
| | | | | Negative | | Positive | |
| | Pix Acc↑ | mAP↑ | mIoU↑ | Pred.↑ | Target↑ | Pred.↓ | Target↓ |
| (i) $\frac{1}{LH}\sum_{l,h} A_{\text{CLS}}^{(l,h)}$ | 67.38 | 80.73 | 44.52 | 49.45 | 49.45 | 23.08 | 23.08 |
| (ii) $\tilde{A}_{\text{CLS}}(c)$ | 68.84 | 81.06 | 47.01 | 50.22 | 50.25 | 22.28 | 22.24 |
| (iii) $\Psi^{(12)}(c)$ | 53.53 | 58.72 | 36.01 | 45.02 | 46.96 | 22.78 | 21.36 |
| PredicAtt$_{12}(c)$ **(Ours)** | **76.85** | **86.24** | **59.22** | **56.16** | **57.94** | **17.00** | **16.06** |

Table 4: **Ablation study on ViT-B/16**. Using only attention maps (lines (i),(ii)) or only prediction maps (line (iii)), leads to weak explainability, yet their integration (Predicatt) yields state-of-the-art results. Note that a weighted average of the attention maps, which is based on their correlation with the prediction map (line (ii)), gives superior results to naively averaging them (line (i)).

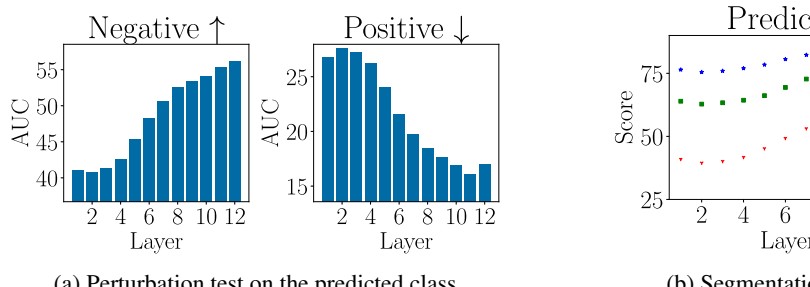

(a) Perturbation test on the predicted class          (b) Segmentation test

Figure 6: **Evaluation of PredicAtt$_i$ per layer $i$.** The optimal score is marked with a circle. Performance generally improves with the use of deeper layers. However, in the segmentation and the positive perturbation tests, the prediction map from layer 11 outperforms that of the last (12th) layer.

optimized during training. Fig. 3 shows the prediction maps of the several last layers. The prediction maps of deeper layers tend to be more semantic, capturing the explained class more coherently.

## 7 CONCLUSION AND LIMITATIONS

We proposed a novel approach for explaining ViT predictions, which is based on a new visualization termed *Prediction Maps*. These maps, together with the well-studied attention maps form integral components of our explainability framework. We showed that the correlation between prediction- and attention-maps reliably indicates the influence of the attention maps on the predictions, and used this to construct a unified explainability map. Our method achieves state-of-the-art results in explainability measures and is significantly faster and more lightweight than the current leading methods. To the best of our knowledge, it is the first gradient-free method that provides class-specific explanations. In the supplementary, we discuss and illustrate the extension of our approach to explaining text models (BERT) and vision-language models (CLIP).

The primary limitation of our approach is its dependence on the classification head accepting tokens of a particular dimension. This assumption does not hold in several architectures, such as in DINO (Caron et al., 2021), where the classification head accepts a concatenation of `[CLS]` tokens from multiple layers, or in Swin (Liu et al., 2021b), where the size of the token embedding varies between layers. In these cases, adjustments to our method are necessary, and we leave them for future work.

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

# A SUPPLEMENTARY MATERIAL

## A.1 PREDICTION MAPS PER LAYER

In this section, we provide quantitative and qualitative results of prediction maps from different layers of ViT-B/16 without combining them with attention maps. Figure 7 shows the perturbation and segmentation tests for the prediction map of each layer, while Fig. 8 shows visual results.

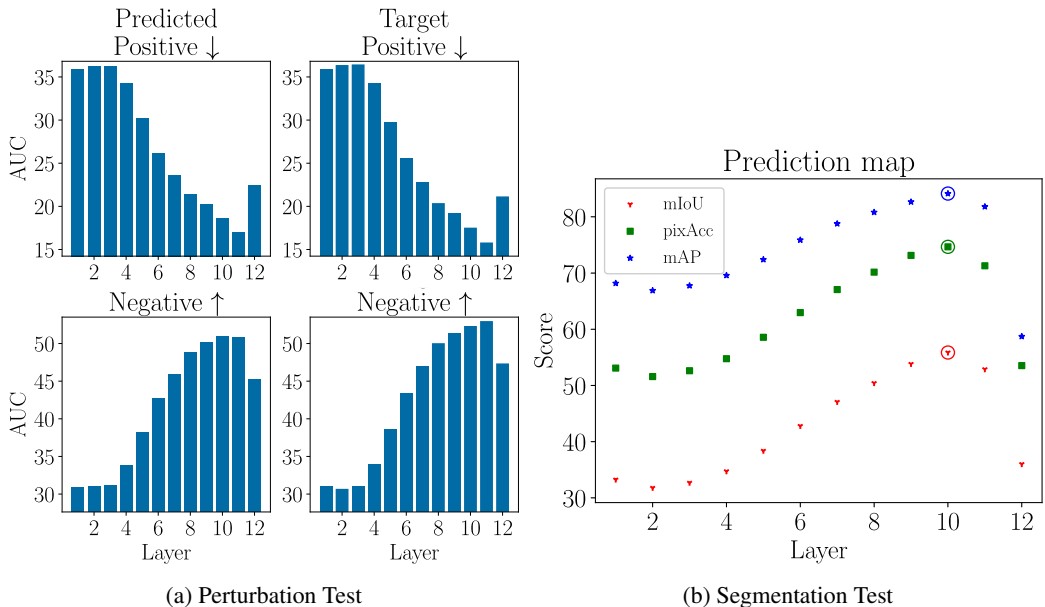

(a) Perturbation Test

(b) Segmentation Test

Figure 7: **Evaluation of prediction map per layer.** The optimal score is marked with a circle. The perturbation tests were run on a subset of the ImageNet validation dataset. Performance generally improves with the use of deeper layers. However, in perturbation tests the prediction map from layer 11 outperforms that of the last (12th) layer, and in the segmentation tests layer 10 outperforms layers 11 and 12.

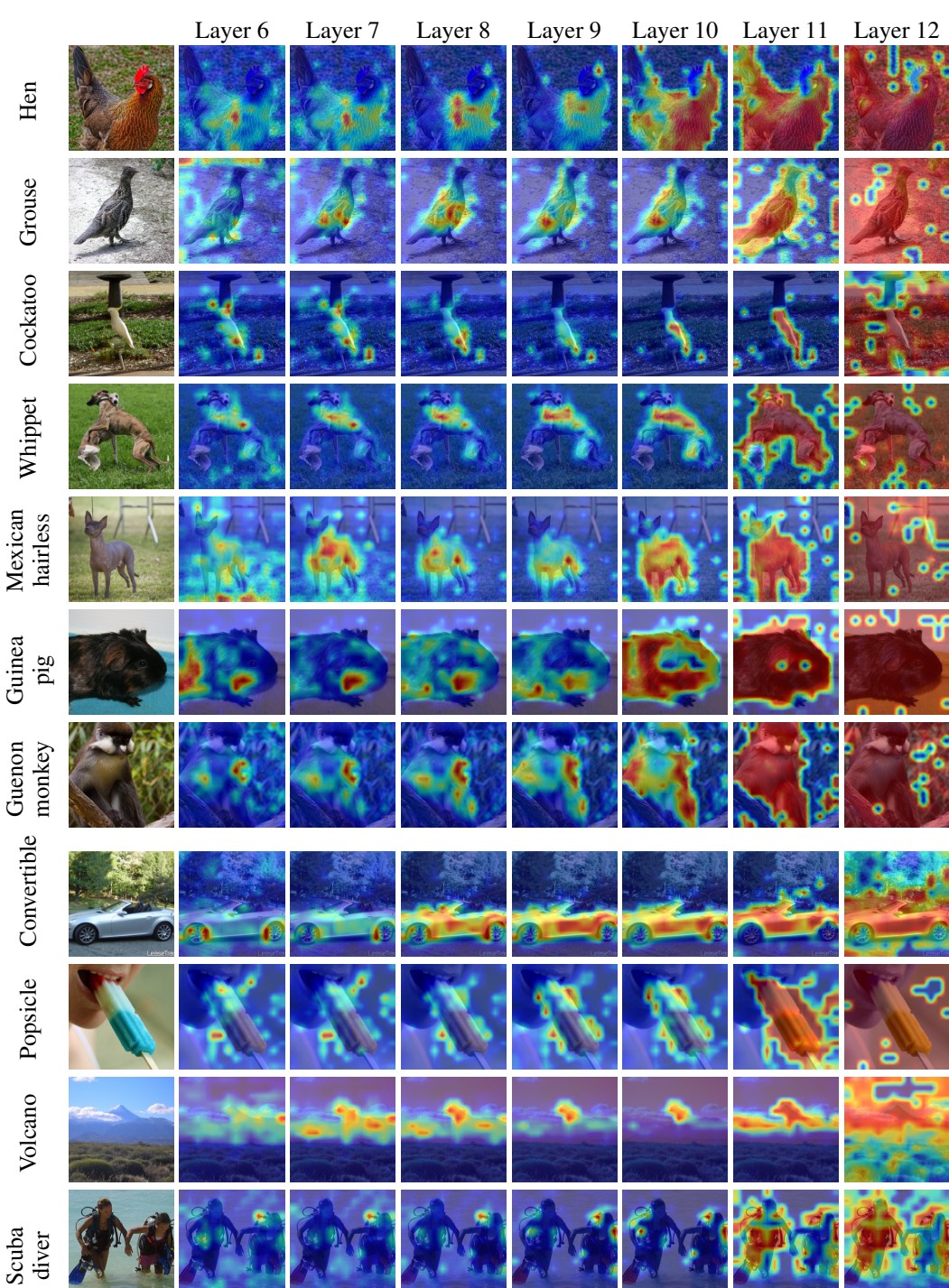

Figure 8: **Prediction map per layer.** Additional examples on images from the ImageNet validation dataset.

## A.2 Alternative Similarity Measure

We now examine an alternative similarity measure for comparing the prediction map with attention maps, which forms the basis for generating the PredicAtt visualization. Specifically, instead of Eq. 4 we use the normalized dot product (correlation coefficient),

$$
\alpha_{l,h} = \frac{\left\langle A_{\text{CLS}}^{(l,h)}, \Psi^{(i)}(c) \right\rangle}{\left\| A_{\text{CLS}}^{(l,h)} \right\| \cdot \left\| \Psi^{(i)}(c) \right\|}. \tag{6}
$$

Table 5 compares the performance of PredicAtt$_{12}$ on the ViT-B/16 model using both the alternative and original similarity measures in segmentation tests. It is evident from the results that the alternative similarity measure is slightly inferior to the original.

| Similarity Measure | Pixel Acc.↑ | mIoU↑ | mAP↑ |
|---|---|---|---|
| Normalized dot-product | 75.60 | 57.76 | 85.85 |
| Dot-product | **76.85** | **59.22** | **86.24** |

Table 5: **Comparison of Similarity Measures.** This table presents the performance results of PredicAtt$_{12}$ on the ViT-B/16 model during segmentation tests, evaluating both the original and alternative similarity measures. The results indicate that the alternative measure, the normalized dot product, demonstrates inferior performance compared to the original method.

## A.3 ADDITIONAL VISUALIZATIONS OF PREDICATT

In Figures 9-15, we provide additional examples of the visualizations generated by our method, PredicAtt, as well as for the similarities between attention and prediction maps at different layers and heads.

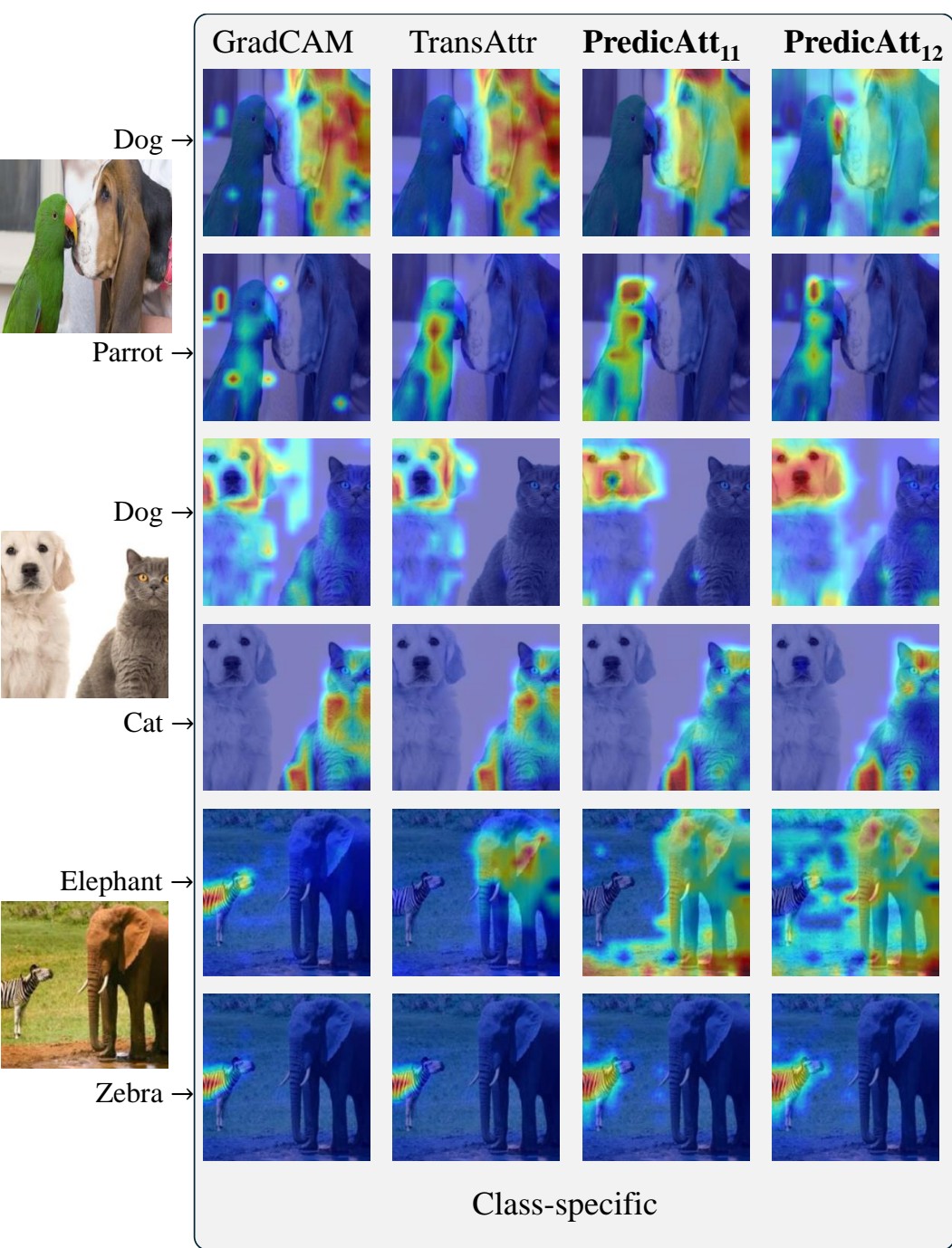

Figure 9: **Class-specific visualizations.**

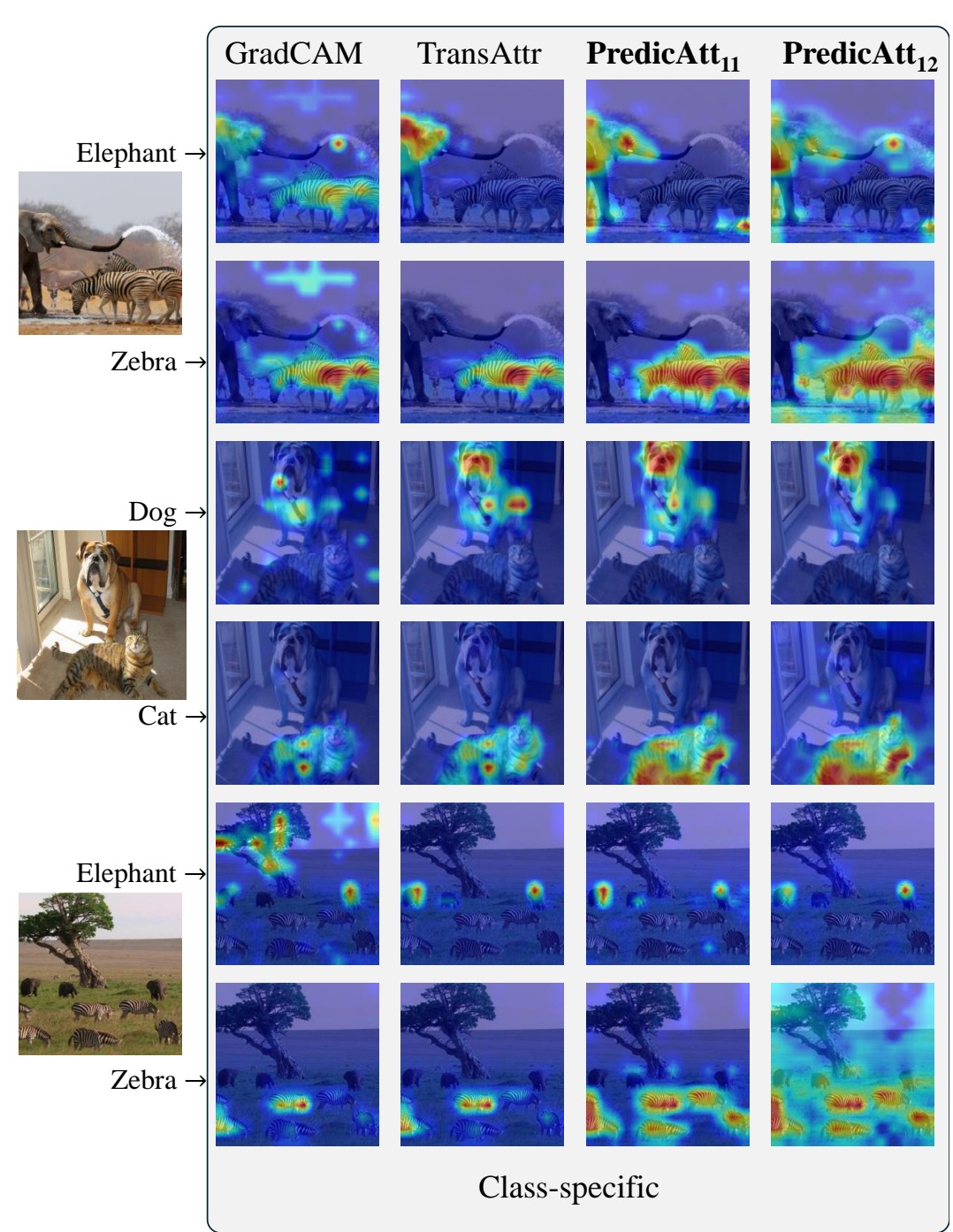

Figure 10: **Class-specific visualizations.**

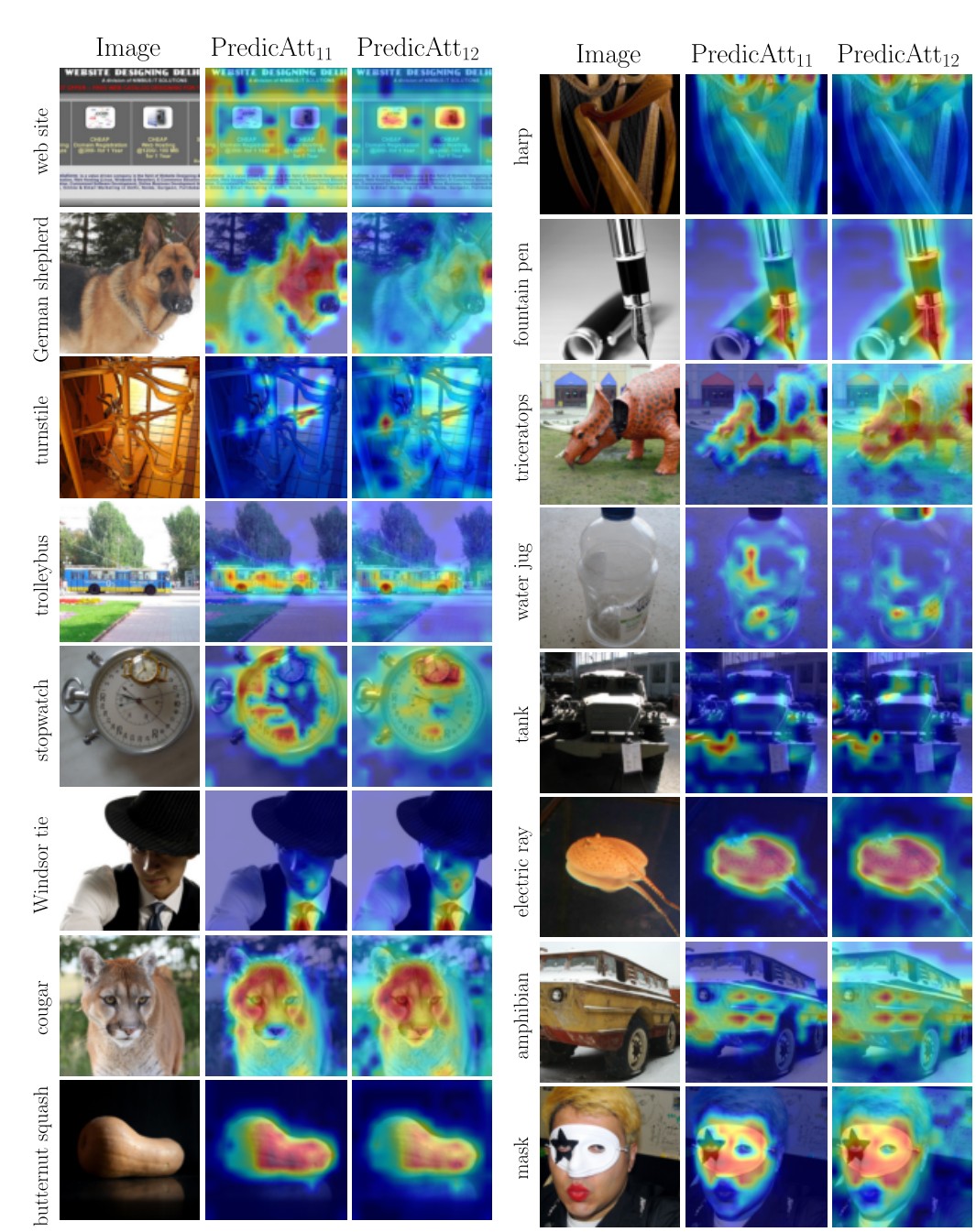

Figure 11: **Sample images from ImageNet validation dataset.** Some images are better explained by PredicAtt$_{11}$, while others by PredicAtt$_{12}$.

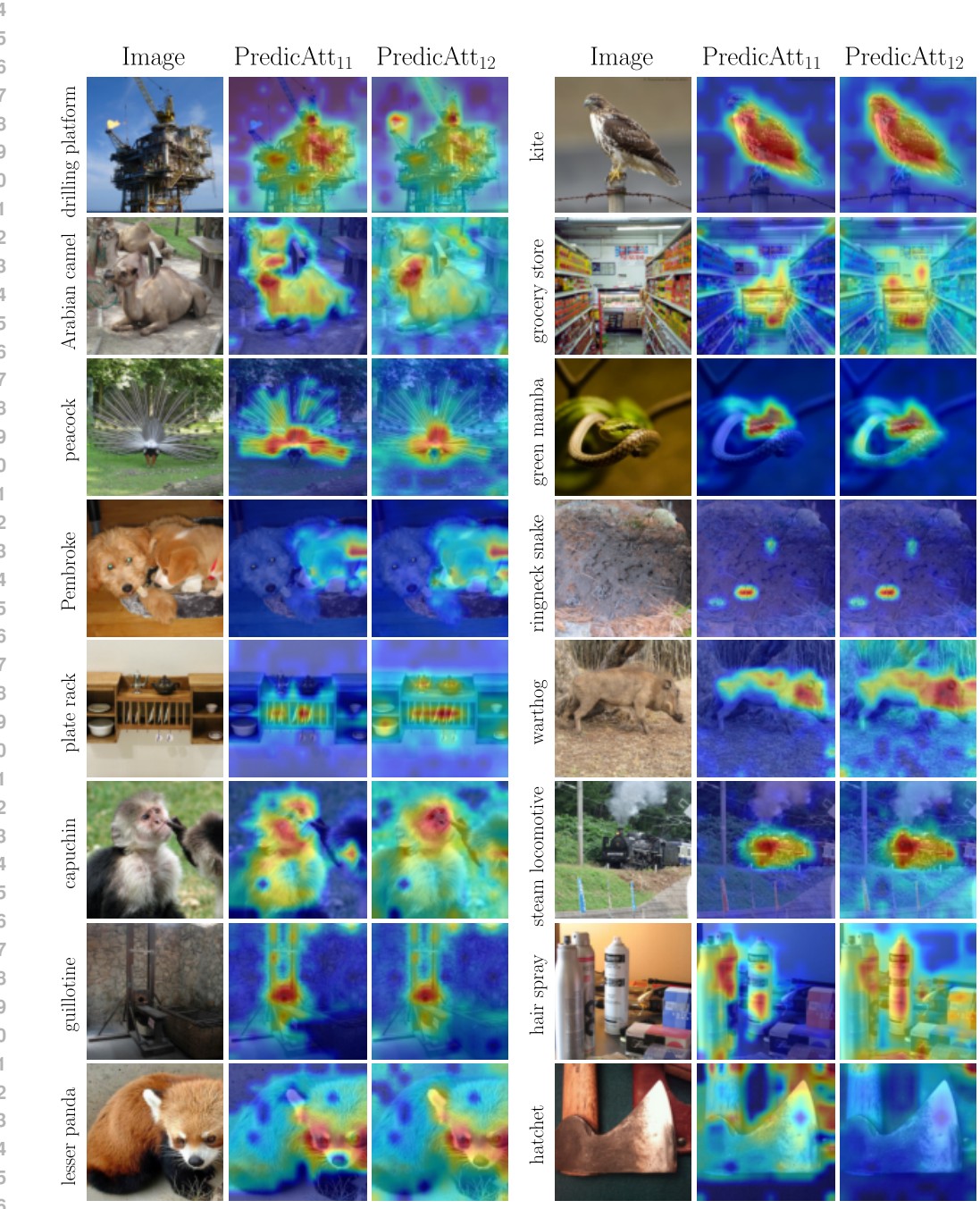

Figure 12: **Sample images from ImageNet validation set.** Some images are better explained by PredicAtt$_{11}$, while others by PredicAtt$_{12}$.

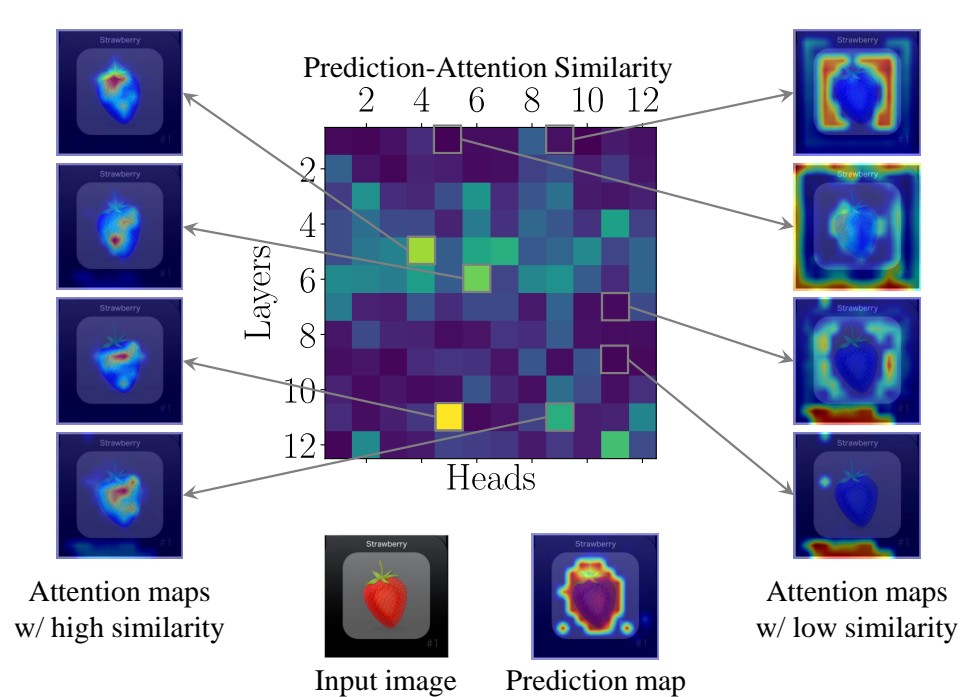

Figure 13: **Per attention map analysis – Strawberry.**

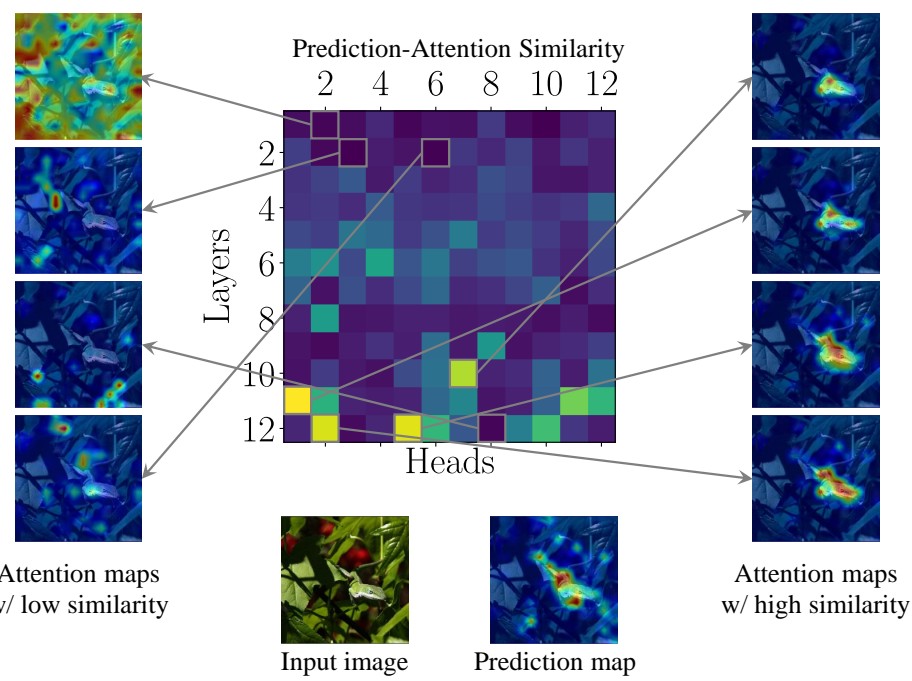

Figure 14: **Per attention map analysis – Chameleon.**

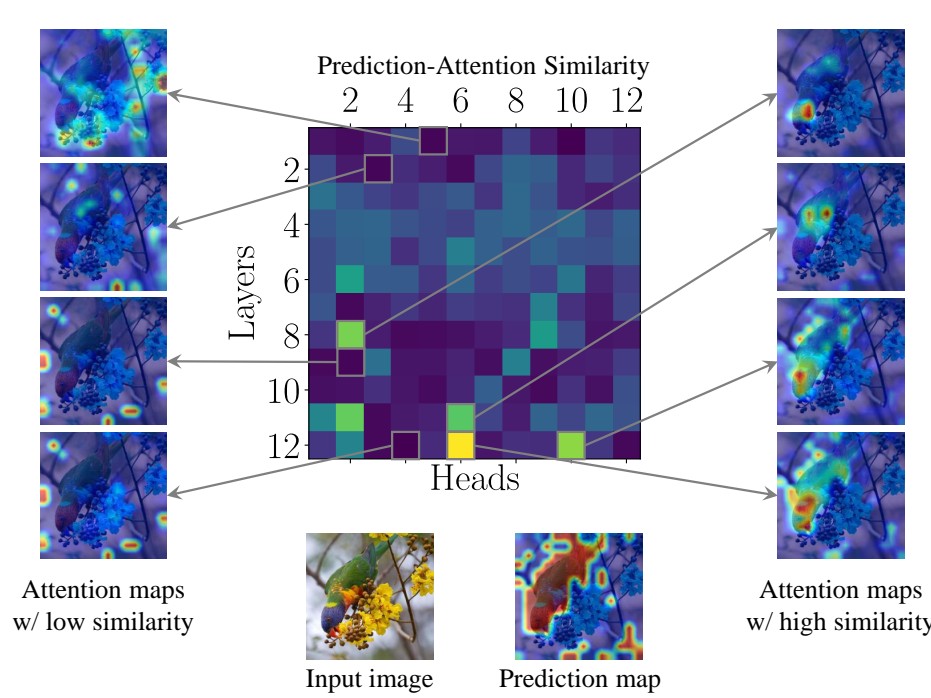

Figure 15: **Per attention map analysis – Lorikeet.**

## A.4   APPLYING PREDICTION MAPS TO TEXT

As discussed in Sec. 7, our method is applicable to any architecture with a classification head that can be fed with tokens other than the CLS token. We focused on image classifiers in order to compare to the existing explainability methods, which also focused on these models. But our method can be applied also to text classifiers with a similar architecture. In that modality, the explainable map would allow highlighting the word-parts in the text, which are most relevant for any desired class in a text-classification task.

Figures 16 and 17 demonstrate the application of prediction maps to a BERT-base (Devlin et al., 2018) classifier model, assuming a maximum token sequence length of 512. Similarly to ViT, a classification token [CLS] is prepended to the input sequence and serves as the input to the classification head. For this illustration, we use a BERT model fine-tuned on the Movie Reviews Dataset (Zaidan et al., 2007), a binary sentiment analysis task. The prediction maps visualize word parts most indicative of either positive or negative sentiment.

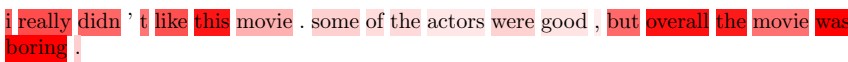

Figure 16: **Prediction Map for Positive Sentiment.** The prediction map highlights word segments associated with positive sentiment.

i really didn ' t like this movie . some of the actors were good , but overall the movie was boring .

Figure 17: **Prediction Map for Negative Sentiment.** The prediction map highlights word segments associated with negative sentiment.

## A.5   ADAPTING PREDICTION MAPS TO CLIP

In this section, we present an adaptation of our method to CLIP (Radford et al., 2021), a vision-language model. Specifically, given an image and a text, we would like to visualize how the CLIP image encoder associates between the text and each region in the image. The CLIP image encoder is not a classification model, and therefore it does not have a classification head. We propose to compute the cosine similarity between the text embedding obtained from the CLIP text encoder and the embedding of each patch token within a given layer in the CLIP image encoder (rather than doing so only for the final embedding at the output of the network). This yields a map that is similar in nature to the prediction maps we obtain for ViT classifiers. An illustration of this adaptation can be seen in Fig. 18. As opposed to our ViT classifier illustrations, here the heatmaps are inverted, with red indicating low similarity values and blue representing high similarity values. This is because we observed that tokens corresponding to the most relevant regions for the prompt rather tend to exhibit the lowest cosine similarity. This inversion occurs consistently across the Transformer layers. We leave the thorough analysis of this phenomenon to future research.

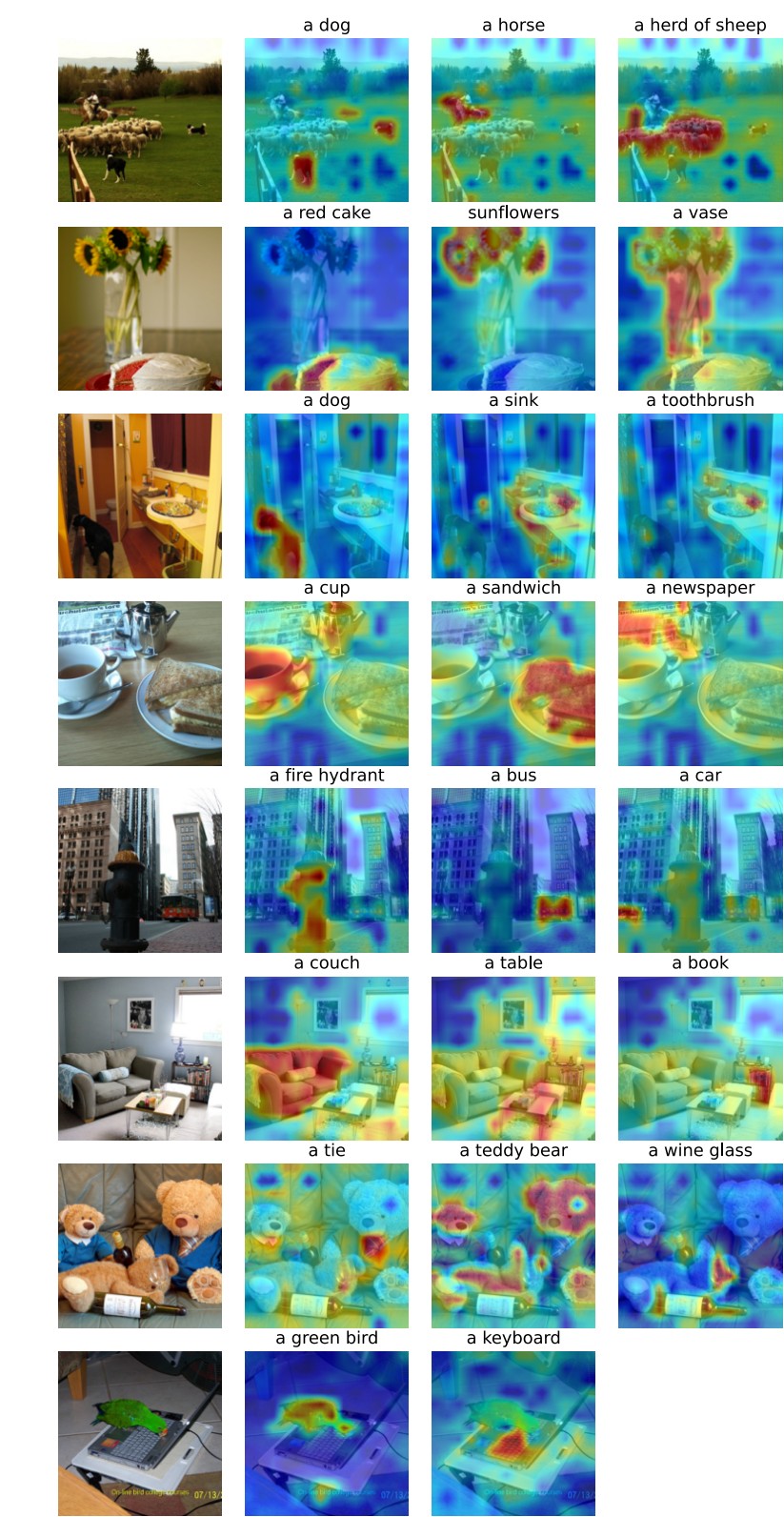

Figure 18: **CLIP explanation maps.** Adaptaion of our method for generating explainable heatmaps for the CLIP ViT-B/16 image encoder model. The patch tokens are extracted from the penultimate layer, with the caption above each heatmap indicating the corresponding prompt used for its generation.

## A.6 ACCURACY, CONFIDENCE AND DISTRIBUTION OF THE PER-LAYER PREDICTIONS

In Figures 19,20,21, we analyze the accuracy, confidence and distribution of the predictions throughout the different layers. As a confidence measure, we utilize the common approach of computing the difference between the logits of the most probable class and the second most probable class, as done in Joseph (2023). As a distribution measure we report the mean entropy of the predictions per layer across the entire dataset. The analysis is based on a random batch of 80 images from the ImageNet validation set. The plots demonstrate the mean value of each metric, calculated separately for the CLS token and the patch tokens across all the sampled images.

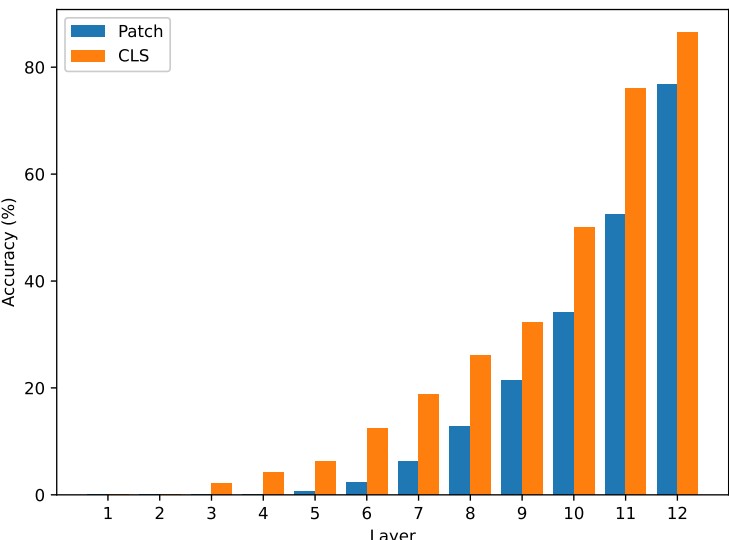

Figure 19: **Accuracy per layer.**

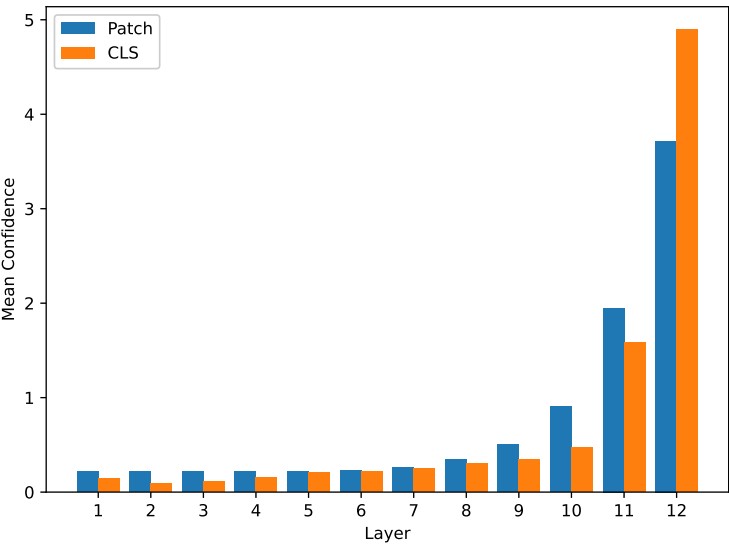

Figure 20: **Mean confidence per layer.**

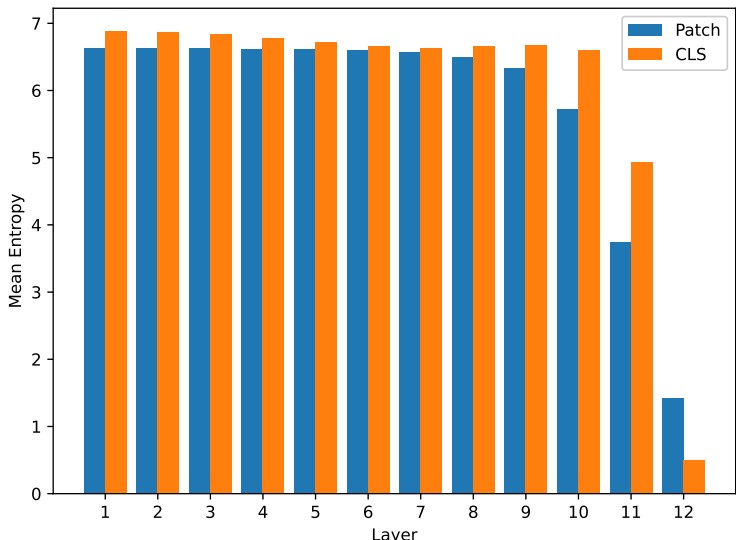

Figure 21: **Mean entropy per layer.**

We measure the similarity of the prediction map of the predicted class from the final layer to the attention heads across all layers. Figure 22 shows a histogram measuring the number of times the most similar attention head came from layer $i$, for every layer in the nework. This analysis, conducted on the ImageNet-Segmentation dataset, reveals an unexpected pattern: while a monotonically increasing trend might be anticipated, the fifth layer emerges as the second most similar layer, surpassed only by the final layer.

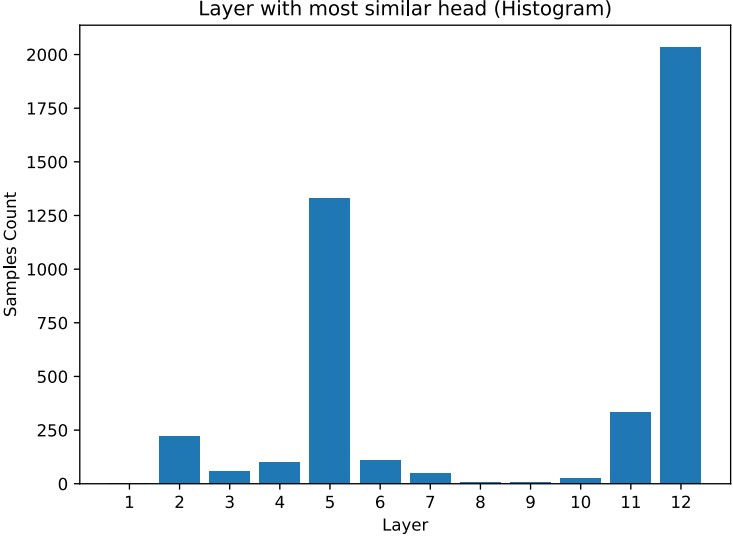

Figure 22: **Layer with the most correlated head.** Histogram of the layers whose attention heads are most similar to the final prediction map on ImageNet-Segmentation. The fifth layer shows notable correlation, second only to the final layer.

### A.7 RESULTS ON SMALLER MODELS

In Tab. 6, we present the results of the perturbation and segmentation tests for ViT-S. Our method still demonstrates a slight improvement over TransAttr across most metrics.

| Method | Perturbation test | | | | Segmentation test | | |
|---|---|---|---|---|---|---|---|
| | Negative | | Positive | | | | |
| | Pred. ↑ | Target ↑ | Pred. ↓ | Target ↓ | pixAcc ↑ | mAP ↑ | mIoU ↑ |
| TransAttr | 53.22 | 53.87 | **14.14** | 13.78 | 80.86 | 86.11 | 63.61 |
| **PredicAtt$_{L-1}$ (Ours)** | **53.57** | **54.87** | 14.24 | **13.55** | **81.26** | **86.17** | **63.94** |
| **PredicAtt$_L$ (Ours)** | 53.22 | 54.61 | 14.64 | 13.96 | 78.13 | 85.12 | 60.29 |

Table 6: **Perturbation and segmentation tests on ViT-S/16.** All methods are evaluated on the ImageNet validation set with the ViT-S/16 model. Bold and underline mark the best and second best scores, respectively. The subscript in our method indicates the layer of the prediction map, where $L$ denotes the total number of layers in the model: 12 for ViT-S/16.

