# OpenReview forum: "From Attention to Prediction Maps: Per-Class Gradient-Free Transformer Explanations"
_ICLR.cc/2025/Conference — Submitted to ICLR 2025_

### Official Review · Reviewer_1rCA · 2024-10-17

**Soundness:** 4
**Presentation:** 4
**Contribution:** 3
**Rating:** 6
**Confidence:** 3

**Summary:**

ViTs are commonly used in computer vision and their interpretation is important. This paper introduces Prediction Maps, which is a novel combination of two interpretability methods. The first method applies the classifier head trained on the CLS token to intermediate patch representations to elicit class information at each patch location and each layer — "the what." The second method leverages the learned attention maps to interpret how spatial information is being moved around the feature map — "the where." Predictions maps are built in two steps. First, class-specific attention maps are computed by taking a weighted sum of attention maps; weights are the similarities between the attention map of a layer-head with the class-specific prediction map. Second, the per-element product is computed between the class-specific weighted attention map and the class-specific prediction map. Prediction Maps can better predict patch importance (table 1), are more useful for segmentation (table 2), and are compute friendly (table 3).

**Strengths:**

- This is an important topic — ViTs are seeing widespread adoption and we need better interpretability methods
- Combining prediction maps and attention maps is novel, and how they are combined is interesting
- Both perturbation and segmentation experiments are convincing to me
- The ablations (table 4) are interesting and important
- I like the BERT and CLIP demonstrations in the appendix
- I'm interested in using this method and I suspect others will be

**Weaknesses:**

I believe the class-wise prediction map is equivalent to the "logit lens" method. Logit lens applies the classifier trained on the final CLS token to intermediate token representations to form token- and layer-wise class predictions. To the best of my knowledge, this was introduced in NLP in 2020 [1] and formally studied in 2023 [2]. The logit lens was used in a ViT interpretability paper [3], another ViT paper [4], and a ViT interpretability repo [5].

Please clarify if your class-wise prediction map is indeed equivalent to the logit lens. If they are equivalent, I still like this paper — it introduces a novel combination of existing interpretability techniques. If they are not equivalent, please clarify.

[1] Blog post: https://www.lesswrong.com/posts/AcKRB8wDpdaN6v6ru/interpreting-gpt-the-logit-lens

[2] Belrose et al. "Eliciting Latent Predictions from Transformers with the Tuned Lens"

[3] Vilas et al. "Analyzing Vision Transformers for Image Classification in Class Embedding Space", NeurIPS 2023

[4] Fuller et al. "LookHere: Vision Transformers with Directed Attention Generalize and Extrapolate", NeurIPS 2024

[5] Repo: https://github.com/soniajoseph/ViT-Prisma

**Questions:**

For the perturbation tests, patches are masked based on their order of importance. Is this importance estimated using each interpretability method — allowing you to compare how well the interpretability methods can predict the classification importance of patches?

**Details Of Ethics Concerns:**

I have no ethics concerns.

---

> ### Author Response · Authors · 2024-11-23
>
> >_``Please clarify if your class-wise prediction map is indeed equivalent to the logit lens. If they are equivalent, I still like this paper — it introduces a novel combination of existing interpretability techniques. If they are not equivalent, please clarify.’’_
>
> Thank you for bringing these works to our attention; we weren’t aware of them. We agree with you that while the prediction map is indeed addressed by the logit lens, our approach of combining it with attention maps remains novel. This combination, as we demonstrate, enhances explainability measures.
> The Logit Lens blog [1] focuses exclusively on the NLP domain, whereas our method adapts it for vision transformers.
> The Tuned Lens paper [2] introduces an optimization step that significantly increases computational overhead compared to prediction maps.
> Vials et al. [3] extend the Logit Lens to ViTs but rely on a gradient-based explanation map.
> Fuller et al. [4] primarily aim to improve positional encoding methods for handling high-resolution images during test time, rather than contributing to explanation techniques.
> The ViT-Prisma repository [5] aligns closely with our prediction maps but visualizes a segmentation map where each patch is marked with an emoji representing the most probable class. In contrast, our visualization operates on a per-class basis.
> Lastly, as you point out, none of these works explore the combination of prediction maps (or their counterparts, such as the logit lens) with attention maps. We agree that this combination provides an insightful perspective on the prediction process, yielding explanations that would not be possible otherwise.
>
> >_``For the perturbation tests, patches are masked based on their order of importance. Is this importance estimated using each interpretability method — allowing you to compare how well the interpretability methods can predict the classification importance of patches?’’_
>
>
> Precisely. We’ll try to better clarify this.

---

> > ### Comment · Reviewer_1rCA · 2024-11-26
> >
> > Thanks for the clarification.
> >
> > As long as this discussion is added to the main paper, I'm happy to keep my score a 6.

---

### Official Review · Reviewer_hdPy · 2024-10-28

**Soundness:** 2
**Presentation:** 1
**Contribution:** 2
**Rating:** 3
**Confidence:** 5

**Summary:**

This paper introduces the concept of Prediction Maps, an explainability method for Vision Transformers (ViTs) that complements traditional attention-based approaches. While attention maps show where the network focuses, Prediction Maps reveal what the network perceives at each location by applying the classification head to individual patch tokens throughout the network's layers (layer probing).
The main contributions are : (1) The introduction of Prediction Maps: A gradient-free, class-specific explainability method that visualizes how each patch token contributes to the different class predictions. (2) PredicAtt: A technique that fuses Prediction Maps with Attention Maps using similarity-based weighting to create comprehensive explanations that capture both what the network sees and where it looks. (3) : State-of-the-art Results: The method achieves superior performance on perturbation and segmentation tests while being computationally more efficient than existing gradient-based approaches and attention-based approaches. (4) By separating "what" from "where" in network predictions, the method enables better diagnosis of failure cases and understanding of how the network processes visual information across layers.

**Strengths:**

The paper introduces a novel approach to explaining Vision Transformer predictions through "Prediction Maps," which repurposes the classification head to analyze individual patch tokens across the layers. The work shows reasonable originality in its gradient-free method for class-specific explanations and its combination of attention and prediction information. The significance lies primarily in providing a computationally efficient tool (gradient-free and class-specific) for model interpretation and debugging.

**Weaknesses:**

1: Line 266, this is not surprising, the core idea of applying a classifier to intermediate representations has been explored before, notably in "LogitLens" [1] and "TunedLens" [2] for NLP which similarly applies the classification head to intermediate tokens to analyze model behavior.


2: The paper should better acknowledge and differentiate from such prior work, especially since the technical approach shares similarities.


3: No theoretical analysis of why patch token predictions are meaningful despite not being trained for this purpose. While Fig. 3 shows some qualitative examples, the paper lacks systematic analysis of how patch token predictions evolve across layers. A quantitative study tracking prediction accuracy, confidence, and class distribution for patch tokens across layers would provide crucial insights into how the network builds its representations.


4: The method requires consistent token dimensions, limiting applicability to architectures like Swin Transformer.


5: Design choices (e.g., dot product similarity in Eq. 4) lack thorough justification (e.g., cosine similarity, which normalizes for magnitude, or KL divergence since we're comparing probability-like distributions).


6: Looking at Tables 1-2, the performance gap between the proposed method and TransAttr appears to shrink with smaller models (ViT-B vs ViT-L). The authors should evaluate their method on even smaller architectures (e.g., ViT-S, ViT-Ti) to verify if the advantages persist. This would help understand if the benefits are truly architectural or simply emerge from scale.


7: The BERT and CLIP extensions feel preliminary and lack quantitative evaluation (e.g .object localization tasks where ground truth bounding boxes are available, activation & pruning tasks [3]).


8: [3] and [4] are highly related related-works, which are not mentioned and compared against.


[1] https://www.lesswrong.com/posts/AcKRB8wDpdaN6v6ru/interpreting-gpt-the-logit-lens


[2] https://arxiv.org/abs/2303.08112


[3] https://arxiv.org/abs/2202.07304


[4] https://arxiv.org/abs/2402.05602

**Questions:**

1. typo - Line 25 : our -> ours.


2. Line 156 - I wouldn't use the word "null", instead a (baseline) image. This acknowledges that while zero/null images are common baseline choices, other choices exist and can be meaningful [1].


3. Line 197: The statement about the [CLS] token needs more precise language. I would suggest revising "It is believed that..." to "The [CLS] token is designed to aggregate information from all other tokens to enable classification based on that token alone." This better reflects the architectural intent.


4. Line 223: The claim about attention maps not being class-specific needs qualification. Add discussion of Multi-class Token Transformer [2] which demonstrates class-specific attention through multiple class tokens. This work shows an alternative approach to obtaining class-discriminative attention maps (while being also gradient-free and class-specific).


5. line 406 - replace "as follows." -> "as follows :"  for proper enumeration.

[1] https://arxiv.org/abs/2310.04821


[2] https://arxiv.org/pdf/2203.02891

---

> ### Author Response · Authors · 2024-11-23
>
> >_``2: The paper should better acknowledge and differentiate from such prior work, especially since the technical approach shares similarities.’’_
>
> We thank the reviewer for referring us to relevant prior work that we had overlooked.
> While the prediction map is indeed addressed by the logit lens, our approach of combining it with attention maps remains novel. This combination, as we demonstrate, enhances explainability measures.
> Moreover, while the Logit Lens blog is confined to the NLP domain, our method extends its applicability to vision transformers, broadening its scope.
> Finally, the Tuned Lens paper incorporates an optimization step that introduces considerable computational overhead, whereas prediction maps offer a more computationally efficient alternative. Moreover, none of the aforementioned works explore the combination of attention and prediction maps, nor do they provide deep insights into the evolution of these mechanisms throughout the different layers and heads of the network. We will add a discussion on these works in the final version, thanks.
>
>
> >_``While Fig. 3 shows some qualitative examples, the paper lacks systematic analysis of how patch token predictions evolve across layers. A quantitative study tracking prediction accuracy, confidence, and class distribution for patch tokens across layers would provide crucial insights into how the network builds its representations.’’_
>
> We thank the reviewer for their suggestion. Kindly note that in Figure 6 (in the main paper) and Figure 7 (in the supplementary) we show evaluation per layer. Additionally, we have updated the PDF and added Appendix A.6 in the supplementary material, where we provide results from the suggested quantitative study. The preliminary results indicate that around the 10th layer, confidence begins to increase while entropy decreases.
>
> In addition, we analyze the layer containing the attention head most similar to the prediction map in Fig. 22. This analysis uncovers an unexpected pattern: instead of a monotonically increasing trend, the fifth layer emerges as the second most similar layer, exceeded only by the final layer. A deeper understanding of these intriguing findings requires further study, which we plan to explore in future work.
>
> Notably, these additional insights are enabled by the novel combination of attention and prediction maps introduced in our work.
>
> >_``5: Design choices (e.g., dot product similarity in Eq. 4) lack thorough justification (e.g., cosine similarity, which normalizes for magnitude, or KL divergence since we're comparing probability-like distributions).’’_
>
> Please note that in Table 5 (in the supplementary) we experiment with another similarity measure (normalized dot-product, a.k.a cosine-similarity), which performs slightly worse than the method we show in the main paper.
>
> >_``6: Looking at Tables 1-2, the performance gap between the proposed method and TransAttr appears to shrink with smaller models (ViT-B vs ViT-L). The authors should evaluate their method on even smaller architectures (e.g., ViT-S, ViT-Ti) to verify if the advantages persist. This would help understand if the benefits are truly architectural or simply emerge from scale.’’_
>
> Thank you for the insightful observation. Please refer to the newly added Table 6 in Appendix A.7 of the updated PDF, where we present the evaluation of our method on ViT-S/16. Even with this smaller architecture, our approach consistently demonstrates some improvement over TransAttr across most metrics.
>
>
> >_``7: The BERT and CLIP extensions feel preliminary and lack quantitative evaluation (e.g .object localization tasks where ground truth bounding boxes are available, activation & pruning tasks [3]).’’_
>
> The illustrations on BERT and CLIP are not meant to be exhaustive. We included these results only to illustrate that our method can potentially be extended to other modalities as well. This is certainly an interesting avenue for future work.
>
>
> >_``8: [3] and [4] are highly related related-works, which are not mentioned and compared against.’’_
>
>
> We’ll mention them in the final version, thanks.
>
> >_``Questions: ...’’_
>
> Thanks for all the corrections and suggested exposition improvements. We incorporated them into the updated version.

---

> ### Comment · Reviewer_hdPy · 2024-11-25
>
> Thank you for your responses. While I appreciate the authors' efforts to address some of my previous comments, I am still largely unsatisfied:
>
> - Revision Clarity:
> I recommend using color-coding (e.g., blue/red text) to highlight the changes made in the revision. This would make it easier to track and evaluate the modifications.
> - BERT and CLIP:
> The authors state that their method is applicable to BERT and CLIP, but this claim lacks empirical validation. For a paper making such claims, I believe it is essential to: (1) Provide quantitative results for these models (2) Include comparative benchmarks.
> - Related Work:
> The authors have not adequately addressed the comparison with [3] and [4], which represent the current state-of-the-art in this field. A thorough comparison with these works is crucial.
> - Performance Improvements: The reported improvement for ViT-Small appears to be marginal. Given that this is one of the main experimental results, the limited performance gain raises questions about the method's practical significance and real-world applicability.
>
> Given the inadequate response to critical concerns and the lack of substantive improvements in the revision, I will keep my score at 3.

---

### Official Review · Reviewer_ELwT · 2024-10-31

**Soundness:** 2
**Presentation:** 2
**Contribution:** 2
**Rating:** 3
**Confidence:** 4

**Summary:**

The paper proposes Prediction Maps as an explainability method for vision transformers. Prediction Maps mainly apply the trained classification head of the network to each patch token at each layer, thus classifying each patch at each step in the pipeline.  The method is evaluated on positive and negative perturbation tests on ImageNet and for segmentation on ImageNet Segmentation.

**Strengths:**

The topic if and how diffusion models are able to capture visual representations is of great interest. The topic is also very timely.

The paper is easy to follow.

**Weaknesses:**

- 1. Is patch classification really explainability?

I understand that the term explainability leaves some room for discussion what falls under it and what not, so using the ISO/IEC TR 29119-11:2020, Sec 3.1.31. here as a base for the following discussion. This considers explainability as the understanding of how the AI-based system came up with a given result. Following this definition I'm not convinced that patch-wise classification actually gives this information, as it is, at the end, patch-wise classification.
Note that patch-wise classification can still do good in localization, and it also makes sense that when you remove patches that carry this class information, the overall classification accuracy drops. But overall, I don't understand how such classification gives us any information on how the system came up with this class. In this case, the conclusion would just be that classification == explanation, so technically, we would not need gradient or attention-based methods, as we could do as well with classification (which would indeed be faster).  Happy to hear what my fellow reviewers are thinking about this.

- 2. Layer-wise patch classification

I understand that the layer-wise patch classification can be used as information on which layer classes are, e.g., constructed (different from the question of how they are constructed). And, given skip-connections, I can understand that class information does not magically appear in the last layer but that some information is carried by the layers before. This might be worth a deeper analysis, but the ablation is a bit shallow in this respect.

**Questions:**

- Does TransAttr refer to (Chefer et al., 2021b)? Please indicate somewhere which publication this refers to.

---

> ### Author Response · Authors · 2024-11-23
>
> >_``But overall, I don't understand how such classification gives us any information on how the system came up with this class.’’_
>
> Thank you for this important discussion. The term “explainability” can indeed mean different things. There is the explainability that interests end-users, which is to explain *what* in the input image triggered the model to output the prediction it did. This type of explainability can assure the user that the model did not base the prediction on e.g. irrelevant parts of the image. There is the explainability that interests engineers, which is to explain *why* the model made certain mistakes (e.g. whether it is because the model attended to a wrong part of the image or because it attended the right part but just didn’t recognize the object correctly there). This type of explainability can help engineers improve models by understanding whether the training set should be enriched with certain types of data, or certain types of augmentations are required. Finally, there is the explainability that interests researchers, which is to explain *how* the model reached the conclusion that it did. This type of explainability can improve our understanding of the inner workings of deep models and can e.g. allow us to devise better architectures.
>
> Our approach (as well as the vast number of other papers on this topic) aims to provides an answer to the two first types of explainability, but it also takes us a step forward in the third type. Specifically, our approach can provide a form of explanation for why the model thinks the class of a given image is what it says it is. In particular, since we examine both prediction maps and attention maps, as illustrated in Fig. 2, our approach helps explain what triggered certain misclassifications - was it because of not attending to the object or because of not recognizing the object? Kindly note that the perturbation and segmentation tests are widely used and considered established measures for these types of explainability. On these measures, our approach outperforms previous methods.
>
> Regarding the third type of explainability, note that our method allows us to probe every token in every layer of the network, visualizing what classes are encoded in that token. This, together with the visualization of the attention maps, helps understand how hypotheses are formed during the forward pass in the network.
>
> >_``Layer-wise patch classification … This might be worth a deeper analysis, but the ablation is a bit shallow in this respect.’’_
>
> In the main text, we provide only one example in Fig. 3, however in Fig. 8 in the supplementary, we provide several more examples. Qualitatively speaking, we can observe that the evolution is somewhat consistent and reaches optimality for layer 11 across a wide range of classes. Quantitatively speaking, we follow the suggestion of Reviewer hdPy to analyze the confidence and distribution of the predictions throughout the different layers. As a confidence measure, we utilize a common approach which computes the difference between the logits of the most confident and the second most confident classes. As a distribution measure we report the mean entropy of the prediction. The preliminary results indicate that around the 10th layer, confidence begins to increase while entropy decreases. For further details, please refer to the newly added Appendix A.6 in the updated PDF.
>
> Additionally, we analyze which layer contains the attention head most similar to the prediction map, as shown in the histogram in Fig. 22. This analysis uncovers an unexpected pattern: instead of a monotonically increasing trend, the fifth layer emerges as the second most similar layer, exceeded only by the final layer and surpassing all other layers by a large margin.
>
> >_``Does TransAttr refer to (Chefer et al., 2021b)? Please indicate somewhere which publication this refers to.’’_
>
> Yes. We clarified this in the updated PDF (line 369), thanks.

---

> ### Comment · Reviewer_ELwT · 2024-11-24
>
> Thanks for the response. I have to admit I'm still not convinced, but perhaps the authors can clarify some additional points:
>
> - "Our approach (as well as the vast number of other papers on this topic) aims to provides an answer to the two first types of explainability"
> If there is a vast number of papers on the "what" side, could you list 3-5 of them? And would it be possible to compare to those directly e.g. in Tab 1 and 2? If not, why not? Just to clarify, I'm looking for evidence that patch based classification (or something similar) is an accepted method in the explainability community. In this case (and if you can provide a better method) I would consider resting my case. But if the main motivation for this paper is to actually establish patch-based classification as a new form of explainability, then I would need a bit more theoretical underpinning beyond "it performs better on x".
>
> - "our approach can provide a form of explanation for why the model thinks the class of a given image is what it says it is"
> I'm inclined to accept the "what", probably less the "why".
> In terms of why, I would rather see an advantage on the gradient based methods. So, probably, what is the theoretical reasoning behind the assumption that the patchwise classification of e.g. layer 6 for a certain ImageNet class explains "why" the network came up with this class? While I accept that this can happen through skip connections, although we are already some layer and transformation away from the final classification layer, it does not give too much information about the why.
>
> - Random remark: It might make sense to look into training-free methods for VL transformers (CLIPSurgery, MASKClip) etc.

---

> > ### Author Response · Authors · 2024-11-25
> >
> > Thank you for the detailed feedback and for giving us the opportunity to clarify our approach further.
> >
> > * Many of the papers we compare ourselves with are explainability methods that aim to address the first two questions. However, they, including the gradient based methods, do not explicitly focus on the third question.
> > As noted by reviewers hdPy and 1rCA, several methods rely on patch-, head-, or layer-wise classification to provide explanations. Representative examples include [1], [2], [3], and [4], which specifically focus on classifying intermediate representations of the network. This supports the claim that patch-level classification is a widely accepted approach in the literature.
> > Among these, TransAttr and GradCAM, two methods we compare against, can also be viewed as performing classification at the token level:
> > TransAttr applies Layer-wise Relevance Propagation (LRP) across the transformer layers for a given class.
> > GradCAM, adapted for transformers, leverages the gradients with respect to attention to weight the attention heads for a given class.
> > Both methods ultimately generate per-patch scores.
> > Nevertheless, none of these methods address the challenge of combining individual classifications into a unified explanation. In agreement with reviewer 1rCA, we believe that the most valuable insights emerge from this combination (e.g., as illustrated in Figures 2 and 4), which represents a novel contribution to the field.
> > Therefore, our primary contribution is not patch-wise classification itself, but rather the integration of attention and prediction maps. To the best of our knowledge, no other work has divided the explanation into these two complementary aspects.
> >
> > * We do not claim that identifying a specific layer provides a complete explanation for why the network selects a given class. Instead, as we previously highlighted, the combination of attention and prediction maps offers valuable insights into this question. For example, consider an image containing multiple classes, such as a cat and a dog. Analyzing the attention map alone may indicate areas of focus but does not clarify whether the network correctly recognizes the objects in the areas of focus. Similarly, patch-wise classification can reveal that certain patches correspond to the cat and others to the dog, but it provides no context about whether the network attends to those regions. Thus, by analyzing each of these maps separately we could not understand why the network came up with the classification. However analyzing both the maps can show that the network tends to attend to a certain class (i.e. the cat). The attention-prediction similarity we presented in Fig. 22 is only a glimpse into a preliminary observation that should be further studied, and does not stand by itself as an explanation for why.
> >
> > * Thanks for referring us to training-free methods for VL transformers.
> >
> >
> > [1] Logit Lens Blog post: https://www.lesswrong.com/posts/AcKRB8wDpdaN6v6ru/interpreting-gpt-the-logit-lens
> >
> > [2] Belrose et al. "Eliciting Latent Predictions from Transformers with the Tuned Lens"
> >
> > [3] Vilas et al. "Analyzing Vision Transformers for Image Classification in Class Embedding Space", NeurIPS 2023
> >
> > [4] Repo: https://github.com/soniajoseph/ViT-Prisma

---

> > > ### Comment · Reviewer_ELwT · 2024-11-30
> > > **Official Comment by Reviewer ELwT**
> > >
> > > Thanks for the response. I'm still not sure in which area this paper falls respectively which might be the closest academic work to compare to. After reviewing the provided references [1-4], my personal feeling is that this method is probably located somewhere between weekly and training-free localization. In this case, I agree with reviewer hdPy and think it would be good to compare it with methods on more advanced VL models. Overall I don't really agree with the explainability part. I there confirm my current rating. As a recommendation for a revision, it might make sense to extend the evaluation to other areas to ground the paper on a broader SoTA.

---

### Official Review · Reviewer_bFAw · 2024-11-03

**Soundness:** 3
**Presentation:** 3
**Contribution:** 4
**Rating:** 6
**Confidence:** 2

**Summary:**

This paper introduces Prediction Maps, a novel explanability method for Vision Transformers (ViTs), specifically for classification tasks. Prediction Maps provide additional insight to attention maps, by providing per-class explanability of what the network "sees" or perceives within each region of an image. The authors achieve this by applying the classification head - originally trained for the class token at the last layer - to every patch token within a given layer. This innovation offers both computational and memory efficiency as it does not rely on gradients or perturbations. Through their empirical studies, they demonstrate that Prediction Maps complemented with attention maps offer impressive explainability performance. They also assert that their method is the first to provide class-specific, gradient-free explainability for ViTs.

**Strengths:**

The paper is quite innovative, presenting a well-explained and distinct approach to ViT prediction explanation. Prediction Maps as an alternative to attention maps are a significant contribution, offering a fresh perspective in examining network interpretations. The method being gradient-free implies it is uniquely efficient in resource consumption, a favorable characteristic compared to other methods requiring significant memory and computational time. Additionally, the method's class-specificity offers a more precise explanation scope compared to other gradient-free methods. The evaluation methods are robust, and the results indicate the method's superior performance in measure tests and resource efficiency. Furthermore, the concept that the technique could be extended to other models, such as BERT or CLIP, is intriguing.

**Weaknesses:**

A notable limitation of this method is that the classification head must accept tokens of a specific dimension, which won't hold for certain architectures like DINO or Swin. This could limit the application spectrum of the method. Also, there is less detail concerning the complete evaluation process and the steps taken to ensure robustness. Further benchmark comparisons with other explanation methods would also be beneficial for the comprehensiveness of the findings.

**Questions:**

1, How would the method be adjusted to handle architectures where the classification head accepts tokens of a varying dimension, such as DINO and Swin?
2. How does the method perform on larger or more complex datasets? Meaning on scale

---

> ### Author Response · Authors · 2024-11-23
>
> >_``A notable limitation of this method is that the classification head must accept tokens of a specific dimension, which won't hold for certain architectures like DINO or Swin.’’_
>
> We agree, the limitations of our approach are acknowledged in the Conclusion and Limitations section. Nonetheless, our method remains relevant due to the widespread use of ViTs for classification tasks, with CLIP as a main example.
> The DINO architecture, for example, involves concatenation of the CLS tokens of several layers. Thus, a straightforward adaptation of our method for DINO could involve concatenating patch tokens in the same way, and applying the classification head on the resulting concatenated vectors. We leave it to future research.

---

> > ### Comment · Reviewer_bFAw · 2024-11-28
> >
> > Thanks for your response. According to the limiations I do not change the score.

---

### Meta-Review · Area_Chair_qpmp · 2024-12-17

**Metareview:**

The submission had mixed reviews, and did not have very strong support. The major concerns are:

1. Requires the classification head to accept tokens of the same dimension as the internal layer [bFAw, hdPy]
2. is patch classification really explainability? Why are patch token predictions meaningful despite not being trained for this purpose? [ELwT, hdPy]
3. analysis on how the information is carried from layer to layer is missing [ELwT, hdPy]
4. missing comparison with XAI methods for VL transformers, CLIPSurgery, MASKClip [ELwT]
5. classifying intermediate representations has been explored before (logit lens) [hdPy, 1rCA]
6. missing justification for some design choices [hdPy]
7. results on BERT and CLIP are preliminary and like quantitative analysis [hdPy]
8. missing comparison to current SoTA XAI methods [hdPY]

After the author response, both ELwT and hdPy were not satisfied (noting issues with Points 4, 7, 8), while
bFAw was still concerned about Point 1.  The AC agrees that more evaluations are needed (4, 7, 8). For VLMs, there is also a recent ICML 2024 work, Grad-ECLIP https://proceedings.mlr.press/v235/zhao24p.html, which can also be applied to ViT classifiers.

ELwT raises a good point that questions how patch token predictions are meaningful explanations -- We can think of these as bottom-up explanations, which examine the extracted features, but do not factor in whether these features are actually used in the prediction (which requires top-down information).  The AC  agrees that the proposed approach is more like weakly or training-free localization/segmentation, and thus additional literature review should be included to discuss the similairties/differences.

The AC agrees with the concerns of the reviewers, and thus recommends reject.

**Additional Comments On Reviewer Discussion:**

see above

---

### Decision · Program_Chairs · 2025-01-22

Reject